# Towards Effective and Interpretable Human-Agent Collaboration in MOBA Games: A Communication Perspective

**Yiming Gao**[1]* **Feiyu Liu**[1]* **Liang Wang**[1] **Zhenjie Lian**[1] **Weixuan Wang**[1]
**Siqin Li**[1] **Xianliang Wang**[1] **Xianhan Zeng**[1] **Rundong Wang**[1] **Jiawei Wang**[1]
**Qiang Fu**[1] **Wei Yang**[1] **Lanxiao Huang**[2] **Wei Liu**[1]
[1]Tencent AI Lab, Shenzhen, China  [2]Tencent TiMi L1 Studio, Chengdu, China
`{yatminggao,marsxliu,enginewang,leolian,waihinwang,gracesqli`
`lemxlwang,ritzeng,rundongwang,donutwang,leonfu,willyang,`
`jackiehuang}@tencent.com;wl2223@columbia.edu`

## Abstract

MOBA games, e.g., *Dota2* and *Honor of Kings*, have been actively used as the testbed for the recent AI research on games, and various AI systems have been developed at the human level so far. However, these AI systems mainly focus on how to compete with humans, less on exploring how to collaborate with humans. To this end, this paper makes the first attempt to investigate human-agent collaboration in MOBA games. In this paper, we propose to enable humans and agents to collaborate through explicit communication by designing an efficient and interpretable **M**eta-**C**ommand **C**ommunication-based framework, dubbed MCC, for accomplishing effective human-agent collaboration in MOBA games. The MCC framework consists of two pivotal modules: 1) an interpretable communication protocol, i.e., the Meta-Command, to bridge the communication gap between humans and agents; 2) a meta-command value estimator, i.e., the Meta-Command Selector, to select a valuable meta-command for each agent to achieve effective human-agent collaboration. Experimental results in *Honor of Kings* demonstrate that MCC agents can collaborate reasonably well with human teammates and even generalize to collaborate with different levels and numbers of human teammates. Videos are available at `https://sites.google.com/view/mcc-demo`.

## 1 Introduction

Games, as the microcosm of real-world problems, have been widely used as testbeds to evaluate the performance of Artificial Intelligence (AI) techniques for decades. Recently, many researchers focus on developing various human-level AI systems for complex games, such as board games like *Go* (Silver et al., 2016; 2017), Real-Time Strategy (RTS) games like *StarCraft 2* (Vinyals et al., 2019), and Multi-player Online Battle Arena (MOBA) games like *Dota 2* (OpenAI et al., 2019). However, these AI systems mainly focus on how to compete instead of collaborating with humans, leaving Human-Agent Collaboration (HAC) in complex environments still to be investigated. In this paper, we study the HAC problem in complex MOBA games (Silva & Chaimowicz, 2017), which is characterized by multi-agent cooperation and competition mechanisms, long time horizons, enormous state-action spaces ($10^{20000}$), and imperfect information (OpenAI et al., 2019; Ye et al., 2020a).

HAC requires the agent to collaborate reasonably with various human partners (Dafoe et al., 2020). One straightforward approach is to improve the generalization of agents, that is, to collaborate with a sufficiently diverse population of teammates during training. Recently, some population-based methods proposed to improve the generalization of agents by constructing a diverse population of partners in different ways, succeeding in video games (Jaderberg et al., 2017; 2019; Carroll et al., 2019; Strouse et al., 2021) and card games (Hu et al., 2020; Andrei et al., 2021). Furthermore, to better evaluate HAC agents, several objective as well as subjective metrics have been proposed (Du et al., 2020; Siu et al., 2021; McKee et al., 2022). However, the policy space in complex MOBA

---

*These authors contributed equally to this work.

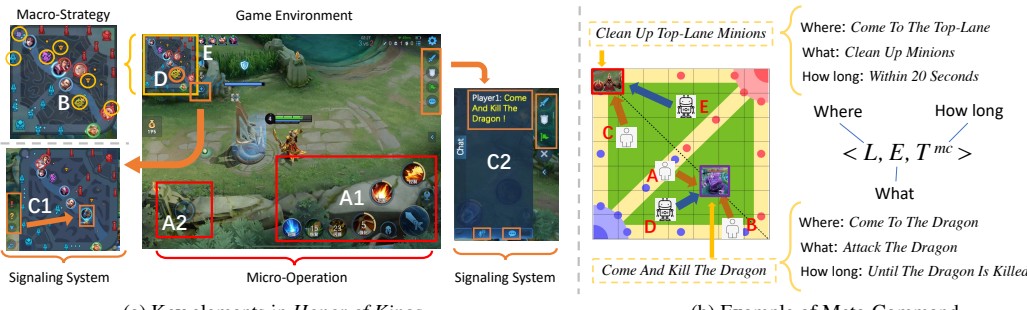

Figure 1: **Introduction of *Honor of Kings*.** (a) Key elements in *Honor of Kings*, including the game environment, micro-operation buttons, examples of macro-strategy, and the signaling system. (b) Example of collaboration via meta-commands. The *Come And Kill The Dragon* is more valuable for humans A and B and agent D to collaborate, while the *Clean Up Top-Lane Minions* is more valuable for human C and agent E to collaborate.

games is enormous (Gao et al., 2021) and requires massive computing resources to build a sufficiently diverse population of agents, posing a big obstacle to the scalability of these methods.

The communication ability to explicitly share information with others is important for agents to collaborate effectively with humans (Dafoe et al., 2020). In Multi-Agent Reinforcement Learning (MARL), communication is often used to improve inter-agent collaboration. Previous work (Sukhbaatar et al., 2016; Foerster et al., 2016; Lazaridou et al., 2016; Peng et al., 2017; Mordatch & Abbeel, 2018; Singh et al., 2018; Das et al., 2019; Wang et al., 2020) mainly focused on exploring communication protocols between multiple agents. Other work (Ghavamzadeh & Mahadevan, 2004; Jiang & Lu, 2018; Kim et al., 2019) proposed to model the value of multi-agent communication for effective collaboration. However, these methods all model communication in latent spaces without considering the human-interpretable *common ground* (Clark & Brennan, 1991; Stalnaker, 2002) or *lingua franca* Kambhampati et al. (2022), making themselves less interpretable to humans. Explicit communication dominated by natural language is often considered in human-robot interaction (Kartoun et al., 2010; Liu et al., 2019; Shafti et al., 2020; Gupta et al., 2021). However, these studies are mainly limited to collaboration between a robot and a human through one-way communication, i.e., humans give robots orders. Therefore, there is still a large room to study RL with the participation of humans.

Success in MOBA games requires subtle individual micro-operations and excellent communication and collaboration among teammates on macro-strategies, i.e., long-term intentions (Wu, 2019; Gao et al., 2021). The micro-operation ability of the existing State-Of-The-Art (SOTA) MOBA agents has exceeded the high-level (top 1%) humans (Ye et al., 2020a). However, these agents' macro-strategies are deterministic and quite different from those of humans (Ye et al., 2020a). Moreover, all existing SOTA MOBA AI systems lack bridges for explicit communication between agents and humans on macro-strategies. These result in the agent's behavior not being understood immediately by humans (Ye et al., 2020a) and not performing well when collaborating with humans (see Section 4.3).

To this end, we propose an efficient and interpretable Meta-Command Communication-based human-agent collaboration framework, dubbed MCC, to achieve effective HAC in MOBA games through explicit communication. First, we design an interpretable communication protocol, i.e., the Meta-Command, as a general representation of macro-strategies to bridge the communication gap between agents and humans. Both macro-strategies sent by humans and messages outputted by agents can be converted into unified meta-commands (see Figure 1(b)). Second, following Gao et al. (2021), we construct a hierarchical model that includes the command encoding network (macro-strategy layer) and the meta-command conditioned action network (micro-action layer), used for agents to generate and execute meta-commands, respectively. Third, we propose a meta-command value estimator, i.e., the Meta-Command Selector, to select the optimal meta-command for each agent to execute. The training process of the MCC agent consists of three phases. We first train the command encoding network to ensure that the agent learns the distribution of meta-commands sent by humans. Afterward, we train the meta-command conditioned action network to ensure that the agent learns to execute meta-commands. Finally, we train the meta-command selector to ensure that the agent learns to select the optimal meta-commands to execute. We train and evaluate the agent in *Honor of Kings* 5v5 mode with a full hero pool (over 100 heroes). Experimental results demonstrate the effectiveness of the MCC framework. In general, our contributions are as follows:

- To the best of our knowledge, we are the first to investigate the HAC problem in MOBA games. We propose the MCC framework to achieve effective HAC in MOBA games.

- We design the Meta-Command to bridge the communication gap between humans and agents. We also propose the Meta-Command Selector to model the agent's value system for meta-commands.

- We introduce the training process of the MCC agent in a typical MOBA game *Honor of Kings* and evaluate it in practical human-agent collaboration tests. Experimental results show that the MCC agent can reasonably collaborate with different levels and numbers of human teammates.

## 2 BACKGROUND

### 2.1 MOBA GAMES

MOBA games have recently received much attention from researchers, especially *Honor of Kings* (Wu, 2019; Ye et al., 2020a;b;c; Gao et al., 2021), one of the most popular MOBA games worldwide. The gameplay is to divide ten players into two camps to compete on the same map. The game environment is shown in Figure 1(a). Each camp competes for resources through individual micro-operations (A1, A2) and team collaboration on macro-strategies (B), and finally wins the game by destroying the enemy's crystal. Players can communicate and collaborate with teammates through the in-game signaling system. Particularly, players can send macro-strategies by dragging signal buttons (C1, C2) to the corresponding locations in the mini-map (D), and these signals display to teammates in the mini-map (E). See Appendix A for detailed game introductions.

### 2.2 HUMAN-AGENT COLLABORATION

We consider an interpretable communicative human-agent collaboration task, which can be extended from the Partially Observable Markov Decision Process (POMDP) and formulated as a tuple $< N, H, \mathbf{S}, \mathbf{A}^N, \mathbf{A}^H, \mathbf{O}, \mathbf{M}, r, P, \gamma >$, where $N$ and $H$ represent the numbers of agents and humans, respectively. $\mathbf{S}$ is the space of global states. $\mathbf{A}^N = \{A_i^N\}_{i=1,\ldots,N}$ and $\mathbf{A}^H = \{A_i^H\}_{i=1,\ldots,H}$ denote the spaces of actions of $N$ agents and $H$ humans, respectively. $\mathbf{O} = \{O_i\}_{i=1,\ldots,N+H}$ denotes the space of observations of $N$ agents and $H$ humans. $\mathbf{M}$ represents the space of interpretable messages. $P : \mathbf{S} \times \mathbf{A}^N \times \mathbf{A}^H \to \mathbf{S}$ and $r : \mathbf{S} \times \mathbf{A}^N \times \mathbf{A}^H \to \mathbb{R}$ denote the shared state transition probability function and reward function of $N$ agents, respectively. Note that $r$ includes both individual rewards and team rewards. $\gamma \in [0, 1)$ denotes the discount factor. For each agent $i$ in state $s_t \in \mathbf{S}$, it receives an observation $o_t^i \in O_i$ and a selected message $c_t^i \in \mathbf{M}$, and then outputs an action $a_t^i = \pi_\theta(o_t^i, c_t^i) \in A_i^N$ and a new message $m_{t+1}^i = \pi_\phi(o_t^i) \in \mathbf{M}$, where $\pi_\theta$ and $\pi_\phi$ are action network and message encoding network, respectively. A message selector $c_t^i = \pi_\omega(o_t^i, C_t)$ is introduced to select a message $c_t^i$ from a message set $C_t = \{m_t^i\}_{i=1,\ldots,N+H} \subset \mathbf{M}$.

We divide the HAC problem in MOBA games into the Human-to-Agent (H2A) and the Agent-to-Human (A2H) scenarios. **The H2A Scenario:** Humans send their macro-strategies as messages to agents, and agents select the optimal one to collaborate with humans based on their value systems. **The A2H Scenario:** Agents send their messages as macro-strategies to humans, and humans select the optimal one to collaborate with agents based on their value systems. The goal of both scenarios is that agents and humans communicate macro-strategies with pre-defined communication protocols and then select valuable macro-strategies for effective collaboration to win the game.

## 3 META-COMMAND COMMUNICATION-BASED FRAMEWORK

In this section, we present the proposed MCC framework in detail. We first briefly describe three key stages of the MCC framework (Section 3.1). Then we introduce its two pivotal modules: 1) an interpretable communication protocol, i.e., the Meta-Command, as a general representation of macro-strategies to bridge the communication gap between agents and humans (Section 3.2); 2) a meta-command value estimator, i.e., the Meta-Command Selector, to model the agent's value system for meta-commands to achieve effective HAC in MOBA games (Section 3.3).

### 3.1 OVERVIEW

The process of the MCC framework consists of three stages: (I) the **Meta-Command Conversion** Stage, (II) the **Meta-Command Communication** Stage, and (III) the **Human-Agent Collaboration**

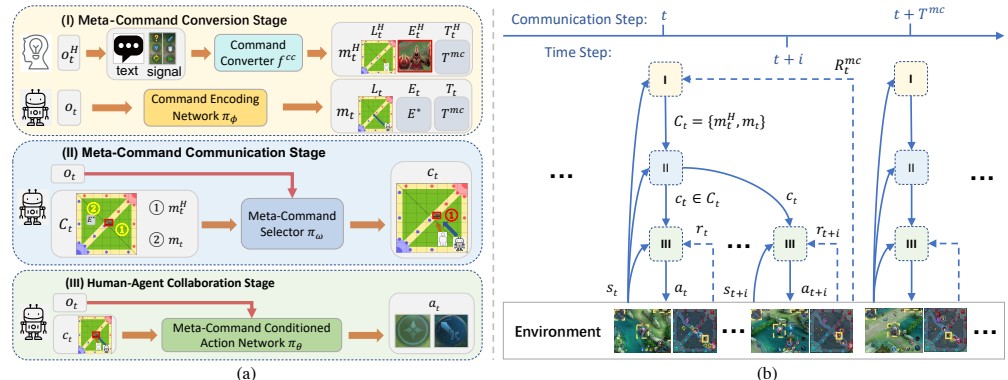

Figure 2: **The MCC framework.** (a) Three key stages of MCC: (I) the meta-command conversion stage, (II) the meta-command communication stage, and (III) the human-agent collaboration stage. (b) The temporal process of MCC. Stage I and II are executed at each communication step. Stage III is executed at each time step.

Stage, as shown in Figure 2. Notably, Stage I and II are executed at each communication step, and Stage III is executed at each time step. At Stage I, the MCC framework converts humans' explicit messages and agents' implicit messages into unified meta-commands $m_t^H, m_t$, respectively, and broadcasts them to all agents and humans. At Stage II, the MCC framework estimates the values of all received meta-commands $C_t$ and selects the optimal one $c_t \in C_t$ for each agent to execute. The selected meta-command will remain unchanged between each two communication steps (e.g. within $[t, t + T^{mc})$ time steps). At stage III, the MCC framework predicts a sequence of actions for each agent to perform based on its selected meta-command $c_t$. In each game, humans and agents collaborate multiple times, that is, execute the three stages multiple times, to win the game.

## 3.2 META-COMMAND

We divide a macro-strategy into three components: where to go, what to do, and how long. For example, a macro-strategy can be *Come And Kill The Dragon*, which consists of *Come To The Dragon Location* (where to go), *Attack The Dragon* (what to do), and *Until The Dragon Is Killed* (how long). Thus, a general representation of macro-strategies, i.e., the Meta-Command, can be formulated as a tuple $< L, E, T^{mc} >$, as shown in Figure 1(b), where $L$ is the *Location* to go, $E$ is the *Event* to do after reaching $L$, and $T^{mc}$ is the *Time Limit* for executing the meta-command.

**Meta-Command Conversion.** To realize bidirectional interpretable human-agent communication, the MCC framework converts humans' explicit messages and agents' implicit messages into unified meta-commands. To achieve the former, we use the Command Converter function $f^{cc}$ (Appendix B.6.1) to extract corresponding location $L^H$ and event $E^H$ from explicit messages sent by humans in the in-game signaling system. To achieve the latter, we use a command encoder network (CEN) $\pi_\phi(m|o)$ to generate $L$ and $E$ based on the agent's observation $o$. The CEN is trained via supervised learning (SL) with the goal of learning the distribution of meta-commands sent by humans (Appendix B.6.2). In MOBA game settings, we use a common location description, i.e., divide $L$ of meta-commands in the map into 144 grids. Since the macro-strategy space is enormous (Gao et al., 2021), customizing corresponding rewards for each specific event to train the agent is not conducive to generalization and is even impossible. Instead, we train a micro-action network to learn to do optimal event $E^*$ at location $L$, just as humans do optimal micro-operations at location $L$ based on their own value systems. We also do not specify a precise $T^{mc}$ for the execution of each specific meta-command. Instead, we set $T^{mc}$ to how long it takes a human to complete a macro-strategy in MOBA games. Usually, 20 seconds corresponds to an 80% completion rate, based on our statistics (Appendix B.6.2). Thus, the MCC framework converts humans' explicit messages into meta-commands $m^H =< L^H, E^H, T^{mc} >$, generates meta-commands $m =< L, E^*, T^{mc} >$ for agents based on their observations (Figure 2(I)), and then broadcasts them to all agents and humans.

**Meta-Command Execution.** After receiving a meta-command candidate set, agents can select one meta-command from it to execute. Note that the MCC framework will replace $E^H$ with $E^*$ when the agent selects a meta-command from humans. We adopt the MCCAN $\pi_\theta(a|o, m)$ for agents to perform actions based on the selected meta-command, as shown in Figure 3(a)(II). The MCCAN is trained via

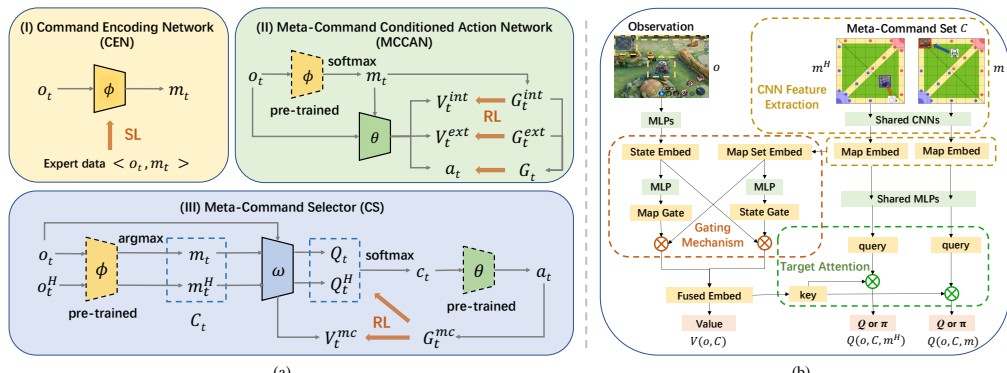

Figure 3: **The training process and model structure of MCC.** (a) The training process is divided into three phases: we first (I) train the CEN via supervised learning (SL), then (II) train the MCCAN via goal-conditioned RL, and finally (III) train the CS via RL. Among them, the dashed box represents the frozen model. (b) The detailed CS model structure, including CNN feature extraction, gating mechanism, target attention module, etc.

self-play RL with the goal of achieving a high completion rate for the meta-commands while ensuring that the win rate is not reduced. To achieve this, we introduce extrinsic rewards $r$ (including individual and team rewards) and intrinsic rewards $r_t^{int}(s_t, m_t, s_{t+1}) = |f^{ce}(s_t) - m_t| - |f^{ce}(s_{t+1}) - m_t|$, where $f^{ce}$ extracts the agent's location from state $s_t$, and $|f^{ce}(s_t) - m_t|$ is the distance between the agent's location and the meta-command's location at time step $t$. Intuitively, the intrinsic rewards are adopted to guide the agent to reach $L$ of the meta-command and stay at $L$ to do some event $E$. The extrinsic rewards are adopted to guide the agent to perform optimal actions to reach $L$ and do optimal event $E^*$ at $L$. Overall, the *optimization objective* is maximizing the expectation over extrinsic and intrinsic discounted total rewards $G_t = \mathbb{E}_{s \sim d_{\pi_\theta}, a \sim \pi_\theta} \left[ \sum_{i=0}^{\infty} \gamma^i r_{t+i} + \alpha \sum_{j=0}^{T^{mc}} \gamma^j r_{t+j}^{int} \right]$, where $d_\pi(s) = \lim_{t \to \infty} P(s_t = s \mid s_0, \pi)$ is the probability when following $\pi$ for $t$ time steps from $s_0$. We use $\alpha$ to weigh the intrinsic and extrinsic rewards.

After training the CEN and the MCCAN, we can achieve HAC by simply setting an agent to randomly select a meta-command derived from humans to execute. However, such collaboration is non-intelligent and can even be a disaster for game victory because agents have no mechanism to model meta-commands' values and cannot choose the optimal meta-command to execute. While humans usually choose the optimal one based on their value systems for achieving effective collaboration to win the game. Thus, we propose a meta-command value estimator to model the agent's value systems for meta-commands, as described in the following subsection.

## 3.3 META-COMMAND SELECTOR

In MOBA games, the same macro-strategy often has different values for different humans in different situations. For example, a macro-strategy can be *Come And Kill The Dragon*, as shown in Figure 1(b). It is more valuable for humans A and B and agent D to collaborate. However, another macro-strategy *Clean Up Top-Lane Minions* is more valuable for human C and agent E than the others. Therefore, agents must select the most valuable meta-command from the received meta-command candidate set $C$ to achieve effective human-agent collaboration. We propose a meta-command value estimator, i.e., the Meta-Command Selector (CS) $\pi_\omega(o, C)$, to estimate the values of all received meta-commands and select the most valuable one for each agent to execute.

**CS Optimization Objective.** Typically, executing a meta-command involves reaching location $L$ and doing event $E$, of which the latter is more important to the value of the meta-command. For example, for the meta-command *Come And Kill The Dragon*, if the event *Kill The Dragon* cannot be done within $T^{mc}$ time steps, then it is pointless to *Come To The Dragon*. Thus, the *optimization objective* of CS is to select the optimal meta-command $m_t^* = \pi_\omega(o_t, C_t)$ for each agent to maximize the expected discounted meta-command execution return $G_t^{mc}$,

$$G_t^{mc} = \mathbb{E}_{s \sim d_{\pi_\theta}, m \sim \pi_\omega, a \sim \pi_\theta} \left[ \sum_{i=0}^{\infty} \gamma_{mc}^i R_{t+i \cdot T^{mc}}^{mc} \right], \quad R_t^{mc} = \underbrace{\sum_{i=0}^{T^L} r_{t+i}}_{(I)} + \beta \underbrace{\sum_{j=T^L}^{T^{mc}} r_{t+j}}_{(II)},$$

where $o_t \in \mathbf{O}$, $C_t$ is the meta-command candidate set in state $s_t$, $\gamma_{mc} \in [0, 1)$ is the discount factor, and $R_t^{mc}$ is a generalized meta-command execution reward function. For $R_t^{mc}$, (I) and (II) are the total extrinsic rewards $r$ before reaching location $L$ and doing event $E$, respectively. $T^L \leq T^{mc}$ is the time for reaching $L$, and $\beta > 1$ is a trade-off parameter representing the relative importance of $E$.

**CS Training Process.** We construct a self-play training environment for CS, where agents can send messages to each other, as shown in Figure 3(a)(III). Specifically, three tricks are adopted to increase the sample efficiency while ensuring efficient exploration. First, each meta-command $m$ is sampled with the argmax rule from the results outputted by the pre-trained CEN. Second, each agent sends its meta-command with a probability $p$ every $T^{mc}$ time steps. Finally, each agent selects the final meta-command $c$ sampled with the softmax rule from its CS output results and hands it over to the pre-trained MCCAN for execution. We use the multi-head value mechanism (Ye et al., 2020a) to model the value of the meta-command execution, which can be formulated as:

$$L^V(\omega) = \mathbb{E}_{O,C} \left[ \sum_{head_k} \|G_k^{mc} - V_\omega^k(O, C)\|_2 \right],$$

where $V_\omega^k(S, C)$ is the value of the $k$-th head. For DQN-based methods (Mnih et al., 2015; Van Hasselt et al., 2016; Wang et al., 2016), the $Q$ loss is:

$$L^Q(\omega) = \mathbb{E}_{O,C,M} \left[ \|G_{total} - Q_\omega^k(O, C, M)\|_2 \right], \quad G_{total} = \sum_{head_k} w_k G_k^{mc},$$

where $w_k$ is the weight of the $k$-th head and $G_k^{mc}$ is the Temporal Difference (TD) estimated value error $R_k^{mc} + \gamma_{mc} V_\omega^k(O', C') - V_\omega^k(O, C)$.

**CS Model Structure.** We design a general network structure for CS, as shown in Figure 3(b). In MOBA games, the meta-commands in adjacent regions have similar values. Thus, we divide the meta-commands in the map into grids, a common location description for MOBA games, and use the shared Convolutional Neural Network (CNN) to extract region-related information to improve the generalization of CS to adjacent meta-commands. Then, the map embeddings of all received meta-commands are integrated into a map set embedding by max-pooling. Besides, we use the gating mechanism (Liu et al., 2021) to fuse the map set embedding and the state embedding of the observation information. Finally, to directly construct the relationship between the observation information and each meta-command, we introduce a target attention module, where the query is the fused embedding and the key is the map embedding of each meta-command. The fused embedding is fed into the subsequent state-action value network $Q(o, C, m)$ and state value network $V(o, C)$ of CS. In this way, we can also easily convert the state-action value network $Q(o, C, m)$ to the policy network $\pi(m|o, C)$. Thus, the CS model structure can be easily applied to the most popular RL algorithms, such as PPO (Schulman et al., 2017), DQN (Mnih et al., 2015), etc.

## 4 EXPERIMENTS

In this section, we evaluate the proposed MCC framework by performing both agent-only and human-agent experiments in *Honor of Kings*. All experiments were conducted in the 5v5 mode with a full hero pool (over 100 heroes, see Appendix A.4).

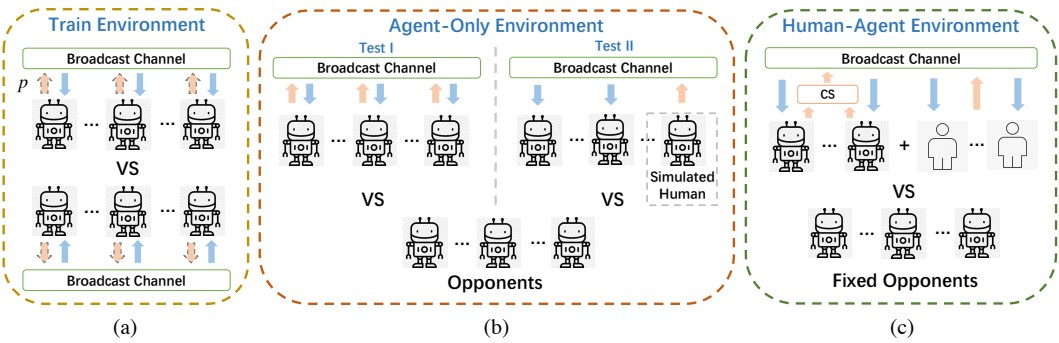

Figure 4: **Communication environments in the experiment.** The orange arrows indicate sending meta-commands, and the blue arrows indicate receiving meta-commands. The dashed line denotes sending meta-commands with probability $p$.

## 4.1 EXPERIMENTAL SETUP

Due to the complexity of MOBA games and limited resources, we train the CEN, the MCCAN, and the CS sequentially instead of training the MCC framework jointly. Specifically, we first train the CEN via SL until it converges for 26 hours using 8 NVIDIA P40 GPUs. The batch size of each GPU is set to 512. Then, we train the MCCAN by fine-tuning the pre-trained WuKong model (Ye et al., 2020a) conditioned on the meta-command sampled from the pre-trained CEN. The MCCAN is trained until it converges for 48 hours using a physical computer cluster with 63,000 CPUs and 560 NVIDIA V100 GPUs. The batch size of each GPU is set to 256. The parameter $\alpha$ is set to 16. After that, we train the CS via self-play until it converges for 24 hours using a physical computer cluster with 70,000 CPUs and 680 NVIDIA V100 GPUs. The batch size of each GPU is set to 256. The parameter $\beta$ is set to 2. Each agent sends a meta-command with a probability $p$ of 0.8 and an interval $T^{mc}$ of 20s, as shown in Figure 4(a). For the entire training process of the MCC framework, the location $L$ of meta-commands in the game map is divided into 144 grids, and the time limit $T^{mc}$ for the meta-command execution is set to 20s. Finally, we obtain the trained MCC agent that can receive meta-commands from other agents and humans and select the most valuable one to execute.

To evaluate the performance of the MCC agent, we conduct both agent-only and human-agent experiments and compare the MCC agent with three different types of agents: the MC-Base agent (agent only executes its own meta-command without communication), the MC-Rand agent (agent randomly selects a meta-command to execute), and the MC-Rule agent (agent selects the nearest meta-command to execute). Note that the MC-Base agent can be considered as a State-Of-The-Art (SOTA) in *Honor of Kings* since it maintains the same capabilities as the WuKong agent (Ye et al., 2020a) (Appendix B.6.3). Results are reported over five random seeds.

## 4.2 AGENT-ONLY COLLABORATION

Directly evaluating agents with humans is expensive, which is not conducive to model selection and iteration. Instead, we built two agent-only testing environments, Test I and Test II, to evaluate agents, as shown in Figure 4(b). Test I is a complex environment where all agent teammates can send and receive meta-commands simultaneously with an interval of 20s. Test I evaluates the agents' performance under complex situations. Test II is a simple environment to simulate most practical game scenarios, where at most one human can send his/her macro-strategy at a time step. Thus, in Test II, only one agent is randomly selected to send its meta-command with an interval of 20s, and the other agents only receive meta-commands. See the detailed experimental results of the CEN and MCCAN in Appendixes B.6.2 and B.6.3, respectively.

**Finding 1: MCC outperforms all baselines.**

We first evaluate the capabilities of the MCC agent and baselines to examine the effectiveness of CS. Figure 5(a) and (b) show the win rates (WRs) of four types of agent teams who play against each other for 600 matches in Test I and Test II, respectively. Figure 5(c) demonstrates the final Elo scores (Coulom, 2008) of these agents. We see that the MCC agent team significantly outperforms the MC-Rand and MC-Rule agent teams. This indicates that compared to selecting meta-commands randomly or by specified rules, the CS can select valuable meta-commands for agents to execute, resulting in effective collaboration. Such collaboration manners in the MCC agent team can even be conducive to winning the game, bringing about 10% WR improvement against the MC-Base agent team. On the contrary, the unreasonable collaboration manners in the MC-Rand and MC-Rule agent teams can hurt performance, leading to significant decreases in the WR against the MC-Base agent team. Note that the MC-Base agent has the same capabilities as the WuKong agent (Ye et al., 2020a), the SOTA in *Honor of Kings*. Overall, the MCC agent achieves the highest Elo scores compared to all baselines in both testing environments, validating the effectiveness of CS. Notably, we also find that the WRs of the MCC agent in Test I and Test II are close, suggesting that the MCC agent can generalize to different numbers of meta-commands. We also investigate the influence of different components, including CNN feature extraction with the gating mechanism (Liu et al., 2021), target attention module, and optimization algorithms on the performance of CS (Appendix B.6.4).

## 4.3 HUMAN-AGENT COLLABORATION

In this section, we conduct an online experiment to evaluate the MCC agent and baselines in collaborating with humans, as shown in Figure 4(c). We contacted the game provider and got a

(a) Win rate (row vs. col) in Test I. (b) Win rate (row vs. col) in Test II. (c) Elo scores of different agents.

Figure 5: **AI performance in the testing environments.** (a) and (b) show the win rate maps of four types of agent teams who play against each other. (c) shows the final Elo scores of these agents.

test authorization. The game provider helped us recruit 30 experienced participants with personal information stripped, including 15 high-level (top1%) and 15 general-level (top30%) participants. We used a within-participant design: *m Human + n Agent* (*mH + nA*) team mode to evaluate the performance of agents teaming up with different numbers of participants, where $m + n = 5$. This design allowed us to evaluate both objective performances as well as subjective preferences.

All participants read detailed guidelines and provided informed consent before the testing. Participants tested 20 matches for the *1H + 4A* team mode. High-level participants tested additional 10 matches for the *2H + 3A* and the *3H + 2A* team modes, respectively. After each test, participants reported their preference over the agent teammates. For fair comparisons, participants were not told the type of their agent teammates. The MC-Base agent team was adopted as the fixed opponent for all tests. To eliminate the effects of collaboration between agents, we prohibit communication between agents. Thus the agents can only communicate with their human teammates. See Appendix C for additional experimental details, including experimental design, result analysis, and ethical review.

**Finding 1: Human-MCC team achieves the highest WR across team modes and human levels.**

We first compare the human-agent team objective performance metrics supported by the MCC agent and baselines, as shown in Table 1. We see that the human-MCC team significantly outperforms all other human-agent teams across different team modes and human levels. This indicates that the MCC agent can generalize to different levels and numbers of human teammates. Note that the SOTA agent can easily beat the high-level human players (Nair, 2019; Chen, 2021). So as the number of participants increases, the WRs of all human-agent teams decrease. Surprisingly, the WR increased significantly when the participants teamed up with the MCC agent. We suspect that the human-MCC team has also achieved effective communication and collaboration on macro-strategies. To verify this, we count the Response Rates (RRs) of agents and participants to the meta-commands sent from their teammates, as shown in Table 2. We find that the RRs of the MCC agents to high-level participants (73.05%) and the high-level participants to the MCC agents (78.5%) are close to the RR of high-level participants themselves (74.91%). This suggests that the CS is close to the value system of high-level humans. Besides, the RRs of participants to the MCC agents (73.43% and 78.5%) are higher than those of the MC-Rand agents (41.07% and 35.69%), indicating that participants collaborated with the MCC agents more often and more effectively.

Table 1: **The win rates (WRs) of different human-agent teams against the fixed opponent.** Each human-agent team consists of different types (MC-Base, MC-Rand, and MCC) of agents and different levels (General and High) and numbers (1H, 2H, and 3H) of participants.

| Agent\Team | 1H + 4A | | 2H + 3A | 3H + 2A |
|---|---|---|---|---|
| | General | High | High | High |
| MC-Base | 23% | 42% | 26% | 8% |
| MC-Rand | 5% | 28% | 18% | 3% |
| MCC | **37%** | **54%** | **39%** | **18%** |

Table 2: **The response rates (RRs) of Receiver to the meta-commands sent by Sender**, including the RRs of agents to participants (H2A scenarios), the RRs of participants to agents (A2H scenarios), and the RRs of participants themselves.

| Sender\Receiver | General | High | MCC |
|---|---|---|---|
| MC-Rand | 41.07% | 35.69% | 34.03% |
| General | 72.34% | - | 61.17% |
| High | - | 74.91% | 73.05% |
| MCC | 73.43% | 78.50% | - |

**Finding 2: Participants prefer MCC over all baselines.**

We then compare the subjective preference metrics, i.e., the Reasonableness of H2A, the Reasonableness of A2H, and the Overall Preference, reported by participants over their agent teammates, as

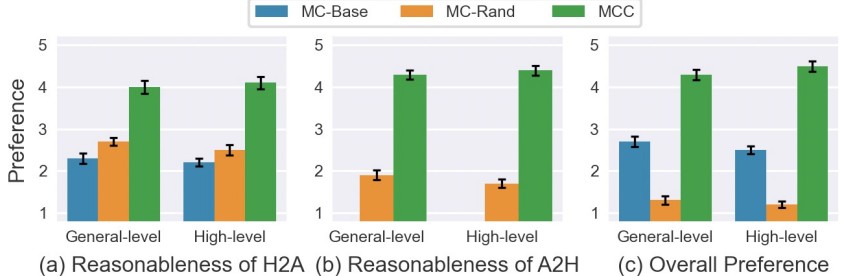

Figure 6: **Participants' preference over their agent teammates.** (a) Reasonableness of H2A: how well the agents respond to your meta-commands. (b) Reasonableness of A2H: how reasonable the meta-commands sent by agents are. (c) Overall Preference: your overall preference for the agent teammates. Participants scored (1: Terrible, 2: Poor, 3: Normal, 4: Good, 5: Perfect) in these metrics after each game test. Error bars represent 95% confidence intervals, calculated over games. See Appendix C.2.3 for detailed wording and scale descriptions.

shown in Figure 6. Participants believed that the MCC agent responded more reasonably and gave the highest score in the Reasonableness of the H2A metric (Figure 6(a)). Besides, participants also believed that the meta-commands sent by the MCC agent are more aligned with their own value system and rated the MCC agent much better than the MC-Rand agent in the Reasonableness of A2H metric (Figure 6(b)). In general, participants were satisfied with the MCC agent over the other agents and gave the highest score in the Overall Preference metric (Figure 6(c)). The results of these subjective preference metrics are also consistent with the results of objective performance metrics.

### 4.4 COLLABORATIVE INTERPRETABILITY ANALYSIS

To better understand how the MCC agents and humans can collaborate effectively and interpretably. We visualize the comparison of CS and high-level participants' value systems on a game scene with three meta-commands existing, as shown in Figure 7. We see that the CS selects the meta-command B for the two heroes in the red dashed box to collaborate, selects the meta-command C for the two heroes in the purple dashed box to collaborate, and selects the meta-command A for the remaining hero to execute alone. The CS selection results are consistent with the ranking results of high-level participants, confirming the effectiveness of the collaboration behaviors between the MCC agents and humans. Such collaboration enhances the human interpretability of agent behavior.

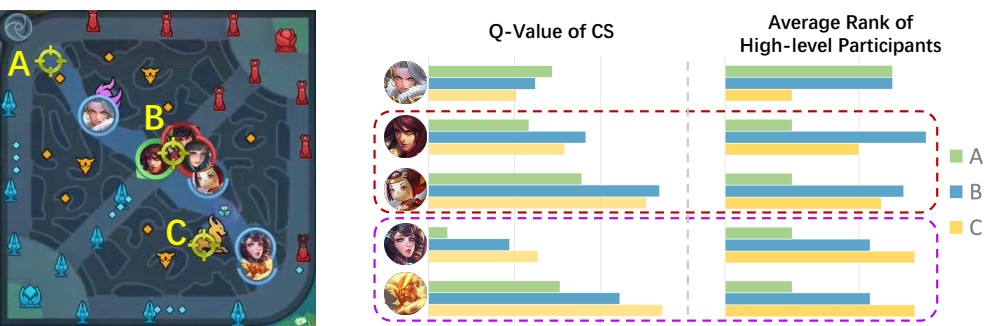

Figure 7: **Case study on the value estimation of CS and average rank of high-level participants.**

### 5 CONCLUSION

In this work, we proposed an efficient and interpretable Meta-Command Communication-based framework, dubbed MCC, to achieve effective human-agent collaboration in MOBA games. To bridge the communication gap between humans and agents, we designed the Meta-Command - a common ground between humans and agents for bidirectional communication. To achieve effective collaboration, we constructed the Meta-Command Selector - a value estimator for agents to select valuable meta-commands to collaborate with humans. Experimental results in *Honor of Kings* demonstrate that MCC significantly outperforms all baselines in both objective performance and subjective preferences, and it could even generalize to different levels and numbers of humans.

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

## A    ENVIRONMENT DETAILS

### A.1    GAME INTRODUCTION

*Honor of Kings* is one of the most popular MOBA games worldwide. The gameplay is to divide ten players into two camps to compete on the same symmetrical map. Players of each camp compete for resources through online confrontation, team collaboration, etc., and finally win the game by destroying the enemy's crystal. The behaviors performed by players in the game can be divided into two categories: macro-strategies and micro-operations. **Macro-strategy** is long-distance scheduling or collaborating with teammates for quick resource competition, such as long-distance support for teammates, collaborating to compete for monster resources, etc. **Micro-operation** is the real-time behavior adopted by each player in various scenarios, such as skill combo release, evading enemy skills, etc. Complicated game maps, diverse hero combinations, diverse equipment combinations, and diverse player tactics make MOBA games extremely complex and exploratory.

### A.2    GAME ENVIRONMENT

Figure 8 shows the UI interface of *Honor of Kings*. For fair comparisons, all experiments in this paper were carried out using a fixed released game engine version (Version 3.73 series) of *Honor of Kings*.

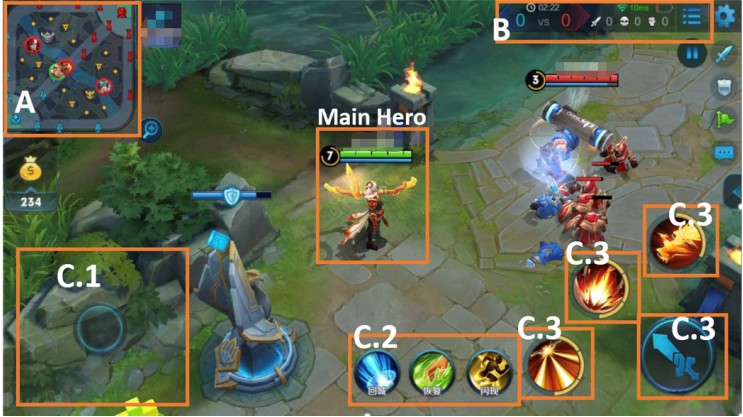

Figure 8: **The UI interface of *Honor of Kings*.** The hero controlled by the player is called *Main Hero*. The player controls the hero's movement through the bottom-left wheel (C.1) and releases the hero's skills through the bottom-right buttons (C.2, C.3). The player can observe the local view via the screen, observe the global view via the top-left mini-map (A), and obtain game states via the top-right dashboard (B).

### A.3    IN-GAME SIGNALING SYSTEM

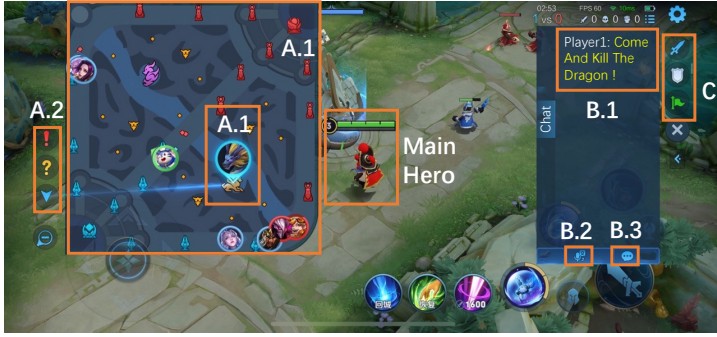 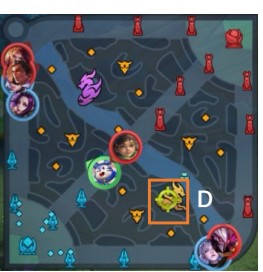

Figure 9: **The in-game signaling system of *Honor of Kings*.** Players can send their macro-strategies by dragging signal buttons (A.2) to the corresponding locations (A.1) in the mini-map. The sent result is displayed as a yellow circle (D). The buttons in C are the convenience signals representing attack, retreat, and assembly, respectively. Voice (B.2) and text (B.1 and B.3) are two other forms of communication.

Figure 9 demonstrates the in-game signaling system of *Honor of Kings*. Players can communicate and collaborate with teammates through the in-game signaling system. In the **Human-Agent Collaboration Test**, humans can send macro-strategies to agents through signals like A in Figure 9, and these signals are displayed to teammates in the form of D. The MCC framework converts these explicit messages, i.e., signals, into meta-commands by the hand-crafted command converter function $f^{cc}$ and broadcasts them to all agent teammates. Moreover, the MCC framework can also convert the meta-commands sent by agents into signals by the inverse of $f^{cc}$ and broadcast them to all human teammates.

Voice (B.2) and text (B.1 and B.3) are two other forms of communication. In the future, we will consider introducing a general meta-command encoding model that can handle all forms of explicit messages (signals, voice, and text).

## A.4 HERO POOL

Table 3 shows the full hero pool and 20 hero pool used in **Experiments**. Each match involves two camps playing against each other, and each camp consists of five randomly picked heroes.

Table 3: Hero pool used in **Experiments**.

| | |
|---|---|
| Full Hero pool | Lian Po, Xiao Qiao, Zhao Yun, Mo Zi, Da Ji, Ying Zheng, Sun Shangxiang, Luban Qihao, Zhuang Zhou, Liu Chan Gao Jianli, A Ke, Zhong Wuyan, Sun Bin, Bian Que, Bai Qi, Mi Yue, Lv Bu, Zhou Yu, Yuan Ge Xia Houdun, Zhen Ji, Cao Cao, Dian Wei, Gongben Wucang, Li Bai, Make Boluo, Di Renjie, Da Mo, Xiang Yu Wu Zetian, Si Mayi, Lao Fuzi, Guan Yu, Diao Chan, An Qila, Cheng Yaojin, Lu Na, Jiang Ziya, Liu Bang Han Xin, Wang Zhaojun, Lan Lingwang, Hua Mulan, Ai Lin, Zhang Liang, Buzhi Huowu, Nake Lulu, Ju Youjing, Ya Se Sun Wukong, Niu Mo, Hou Yi, Liu Bei, Zhang Fei, Li Yuanfang, Yu Ji, Zhong Kui, Yang Yuhuan, Chengji Sihan Yang Jian, Nv Wa, Ne Zha, Ganjiang Moye, Ya Dianna, Cai Wenji, Taiyi Zhenren, Donghuang Taiyi, Gui Guzi, Zhu Geliang Da Qiao, Huang Zhong, Kai, Su Lie, Baili Xuance, Baili Shouyue, Yi Xing, Meng Qi, Gong Sunli, Shen Mengxi Ming Shiyin, Pei Qinhu, Kuang Tie, Mi Laidi, Yao, Yun Zhongjun, Li Xin, Jia Luo, Dun Shan, Sun Ce Zhu Bajie, Shangguan Waner, Ma Chao, Dong Fangyao, Xi Shi, Meng Ya, Luban Dashi, Pan Gu, Chang E, Meng Tian Jing, A Guduo, Xia Luote, Lan, Sikong Zhen, Erin, Yun ying, Jin Chan, Fei, Sang Qi |
| 20 Hero Pool | Jing, Pan Gu, Zhao Yun, Ju Youjing, Donghuang Taiyi, Zhang Fei, Gui Guzi, Da Qiao, Sun Shangxiang, Luban Qihao, Chengji Sihan, Huang Zhong, Zhuang Zhou, Lian Po, Liu Bang, Zhong Wuyan, Yi Xing, Zhou Yu, Xi Shi, Zhang Liang |

## B FRAMEWORK DETAILS

## B.1 INFRASTRUCTURE DESIGN

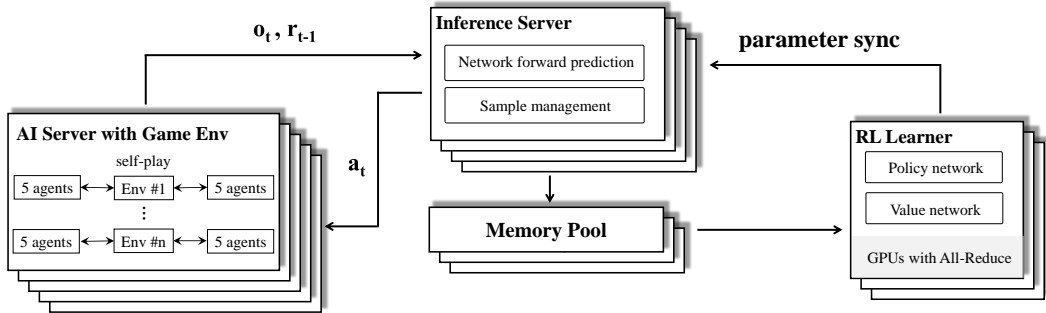

Figure 10: The designed infrastructure.

Figure 10 shows the infrastructure of the training system (Ye et al., 2020a), which consists of four key components: AI Server, Inference Server, RL Learner, and Memory Pool. The AI Server (the actor) covers the interaction logic between the agents and the environment. The Inference Server is used for the centralized batch inference on the GPU side. The RL Learner (the learner) is a distributed training environment for RL models. And the Memory Pool is used for storing the experience, implemented as a memory-efficient circular queue.

Training complex game AI systems often require a large number of computing resources, such as AlphaGo Lee Sedol (280 GPUs), OpenAI Five Final (1920 GPUs), and AlphaStar Final (3072 TPUv3

cores). We also use hundreds of GPUs for training the agents. A worthy research direction is to improve resource utilization using fewer computing resources.

## B.2 REWARD DESIGN

Table 4 demonstrates the details of the designed environment reward.

Table 4: The details of the environment reward.

| Head | Reward Item | Weight | Type | Description |
|------|-------------|--------|------|-------------|
| Farming Related | Gold | 0.005 | Dense | The gold gained. |
| | Experience | 0.001 | Dense | The experience gained. |
| | Mana | 0.05 | Dense | The rate of mana (to the fourth power). |
| | No-op | -0.00001 | Dense | Stop and do nothing. |
| | Attack monster | 0.1 | Sparse | Attack monster. |
| KDA Related | Kill | 1 | Sparse | Kill a enemy hero. |
| | Death | -1 | Sparse | Being killed. |
| | Assist | 1 | Sparse | Assists. |
| | Tyrant buff | 1 | Sparse | Get buff of killing tyrant, dark tyrant, storm tyrant. |
| | Overlord buff | 1.5 | Sparse | Get buff of killing the overlord. |
| | Expose invisible enemy | 0.3 | Sparse | Get visions of enemy heroes. |
| | Last hit | 0.2 | Sparse | Last hitting an enemy minion. |
| Damage Related | Health point | 3 | Dense | The health point of the hero (to the fourth power). |
| | Hurt to hero | 0.3 | Sparse | Attack enemy heroes. |
| Pushing Related | Attack turrets | 1 | Sparse | Attack turrets. |
| | Attack crystal | 1 | Sparse | Attack enemy home base. |
| Win/Lose Related | Destroy home base | 2.5 | Sparse | Destroy enemy home base. |

## B.3 FEATURE DESIGN

### B.3.1 CEN

See Table 5.

Table 5: Feature details of CEN.

| Feature Class | Field | Description | Dimension | Type |
|---------------|-------|-------------|-----------|------|
| **1. Unit feature** | Scalar | Includes heroes, minions, monsters, and turrets | 3946 | |
| Heroes | Status | Current HP, mana, speed, level, gold, KDA, and magical attack and defense, etc. | 1562 | (one-hot, normalized float) |
| | Position | Current 2D coordinates | 20 | (normalized float) |
| Minions | Status | Current HP, speed, visibility, killing income, etc. | 920 | (one-hot, normalized float) |
| | Position | Current 2D coordinates | 80 | (normalized float) |
| Monsters | Status | Current HP, speed, visibility, killing income, etc. | 728 | (one-hot, normalized float) |
| | Position | Current 2D coordinates | 56 | (normalized float) |
| Turrets | Status | Current HP, locked targets, attack speed, etc. | 540 | (one-hot, normalized float) |
| | Position | Current 2D coordinates | 40 | (normalized float) |
| **2. In-game stats feature** | Scalar | Real-time statistics of the game | 104 | |
| Static statistics | Time | Current game time | 57 | (one-hot) |
| | Camp | Types of two camps | 1 | (one-hot) |
| | Alive heroes | Number of alive heroes of two camps | 10 | (one-hot) |
| | Kill | Number of Kill of each camp (Segment representation) | 6 | (one-hot) |
| | Alive turrets | Number of alive turrets of each camp | 8 | (one-hot) |
| Comparative statistics | Alive heroes diff | Alive heroes difference between two camps | 11 | (one-hot) |
| | Kill diff | Kill difference between two camps | 5 | (one-hot) |
| | Alive turrets diff | Alive turrets difference between two camps | 6 | (one-hot) |

### B.3.2   MCCAN

See Table 6.

Table 6: Feature details of MCCAN.

| Feature Class | Field | Description | Dimension | Type |
|---|---|---|---|---|
| **1. Unit feature** | Scalar | Includes heroes, minions, monsters, and turrets | 8599 | |
| Heroes | Status | Current HP, mana, speed, level, gold, KDA, buff, bad states, orientation, visibility, etc. | 1842 | (one-hot, normalized float) |
| | Position | Current 2D coordinates | 20 | (normalized float) |
| | Attribute | Is main hero or not, hero ID, camp (team), job, physical attack and defense, magical attack and defense, etc. | 1330 | (one-hot, normalized float) |
| | Skills | Skill 1 to Skill N's cool down time, usability, level, range, buff effects, bad effects, etc. | 2095 | (one-hot, normalized float) |
| | Item | Current item lists | 60 | (one-hot) |
| Minions | Status | Current HP, speed, visibility, killing income, etc. | 1160 | (one-hot, normalized float) |
| | Position | Current 2D coordinates | 80 | (normalized float) |
| | Attribute | Camp (team) | 80 | (one-hot) |
| | Type | Type of minions (melee creep, ranged creep, siege creep, super creep, etc.) | 200 | (one-hot) |
| Monsters | Status | Current HP, speed, visibility, killing income, etc. | 868 | (one-hot, normalized float) |
| | Position | Current 2D coordinates | 56 | (normalized float) |
| | Type | Type of monsters (normal, blue, red, tyrant, overlord, etc.) | 168 | (one-hot) |
| Turrets | Status | Current HP, locked targets, attack speed, etc. | 520 | (one-hot, normalized float) |
| | Position | Current 2D coordinates | 40 | (normalized float) |
| | Type | Type of turrets (tower, high tower, crystal, etc.) | 80 | (one-hot) |
| **2. In-game stats feature** | Scalar | Real-time statistics of the game | 68 | |
| Static statistics | Time | Current game time | 5 | (one-hot) |
| | Gold | Golds of two camps | 12 | (normalized float) |
| | Alive heroes | Number of alive heroes of two camps | 10 | (one-hot) |
| | Kill | Kill number of each camp (Segment representation) | 6 | (one-hot) |
| | Alive turrets | Number of alive turrets of two camps | 8 | (one-hot) |
| Comparative statistics | Gold diff | Gold difference between two camps (Segment representation) | 5 | (one-hot) |
| | Alive heroes diff | Alive heroes difference between two camps | 11 | (one-hot) |
| | Kill diff | Kill difference between two camps | 5 | (one-hot) |
| | Alive turrets diff | Alive turrets difference between two camps | 6 | (one-hot) |
| **3. Invisible opponent information** | Scalar | Invisible information used for the value net | 560 | |
| Opponent heroes | Position | Current 2D coordinates, distances, etc. | 120 | (normalized float) |
| NPC | Position | Current 2D coordinates of all non-player characters, including minions, monsters, and turrets | 440 | (normalized float) |
| **4. Spatial feature** | Spatial | 2D image-like, extracted in channels for convolution | 6x17x17 | |
| Skills | Region | Potential damage regions of ally and enemy skills | 2x17x17 | |
| | Bullet | Bullets of ally and enemy skills | 2x17x17 | |
| Obstacles | Region | Forbidden region for heroes to move | 1x17x17 | |
| Bushes | Region | Bush region for heroes to hide | 1x17x17 | |
| **5. Meta-Command feature** | Spatial | Flattened Meta-Command | 144 | (one-hot) |

### B.3.3 CS

See Table 7.

Table 7: Feature details of CS.

| Feature Class | Field | Description | Dimension | Type |
|---|---|---|---|---|
| **1. Unit feature** | Scalar | Includes heroes, minions, monsters, and turrets | 3946 | |
| Heroes | Status | Current HP, mana, speed, level, gold, KDA, and magical attack and defense, etc. | 1562 | (one-hot, normalized float) |
| | Position | Current 2D coordinates | 20 | (normalized float) |
| Minions | Status | Current HP, speed, visibility, killing income, etc. | 920 | (one-hot, normalized float) |
| | Position | Current 2D coordinates | 80 | (normalized float) |
| Monsters | Status | Current HP, speed, visibility, killing income, etc. | 728 | (one-hot, normalized float) |
| | Position | Current 2D coordinates | 56 | (normalized float) |
| Turrets | Status | Current HP, locked targets, attack speed, etc. | 540 | (one-hot, normalized float) |
| | Position | Current 2D coordinates | 40 | (normalized float) |
| **2. In-game stats feature** | Scalar | Real-time statistics of the game | 104 | |
| Static statistics | Time | Current game time | 57 | (one-hot) |
| | Camp | Types of two camps | 1 | (one-hot) |
| | Alive heroes | Number of alive heroes of two camps | 10 | (one-hot) |
| | Kill | Kill number of each camp | 6 | (one-hot) |
| | Alive turrets | Number of alive turrets of two camps | 8 | (one-hot) |
| Comparative statistics | Alive heroes diff | Alive heroes difference between two camps | 11 | (one-hot) |
| | Kill diff | Kill difference between two camps | 5 | (one-hot) |
| | Alive turrets diff | Alive turrets difference between two camps | 6 | (one-hot) |
| **3. Invisible opponent information** | Scalar | Invisible information used for the value net | 560 | |
| Opponent heroes | Position | Current 2D coordinates, distances, etc. | 120 | (normalized float) |
| NPC | Position | Current 2D coordinates of all non-player characters, including minions, monsters, and turrets | 440 | (normalized float) |
| **4. Meta-Command feature** | Spatial | 2D image-like, extracted in channels for convolution | 5x12x12 | |
| Meta-Commands | Spatial | All received Meta-Commands in the team | 5x12x12 | (one-hot) |

### B.4 AGENT ACTION

Table 8 shows the action space of agents.

Table 8: The action space of agents.

| Action | Detail | Description |
|---|---|---|
| What | Illegal action | Placeholder. |
| | None action | Executing nothing or stopping continuous action. |
| | Move | Moving to a certain direction determined by move x and move y. |
| | Normal Attack | Executing normal attack to an enemy unit. |
| | Skill1 | Executing the first skill. |
| | Skill2 | Executing the second skill. |
| | Skill3 | Executing the third skill. |
| | Skill4 | Executing the fourth skill (only a few heroes have Skill4). |
| | Summoner ability | An additional skill choosing before the game begins (10 to choose). |
| | Return home(Recall) | Returning to spring, should be continuously executed. |
| | Item skill | Some items can enable an additional skill to player's hero. |
| | Restore | Blood recovering continuously in 10s, can be disturbed. |
| | Collaborative skill | Skill given by special ally heroes. |
| How | Move X | The x-axis offset of moving direction. |
| | Move Y | The y-axis offset of moving direction. |
| | Skill X | The x-axis offset of a skill. |
| | Skill Y | The y-axis offset of a skill. |
| Who | Target unit | The game unit(s) chosen to attack. |

## B.5 NETWORK ARCHITECTURE

### B.5.1 CEN

Figure 11 shows the detailed model structure of CEN. The CEN network extracts game stats features and serval unit features from the observation $o$. Each unit feature is encoded by shared MLP layers to obtain serval unit embeddings, and the game stats features are encoded by MLP layers to obtain the game stats embedding. Finally, we concatenate the unit embeddings and the game stats embedding, and output the probability distribution of the meta-commands. The outputted meta-command indicates the macro-strategy for future $T^{mc}$ time steps.

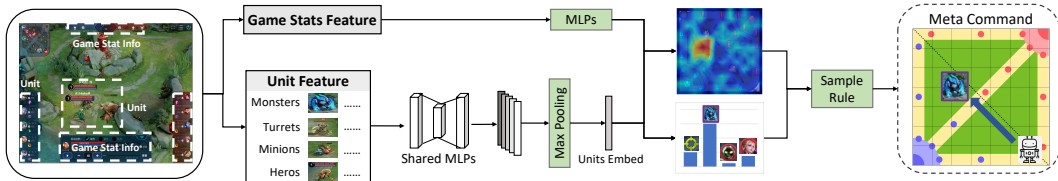

Figure 11: The CEN model structure.

### B.5.2 MCCAN

Figure 12 shows the detailed model structure of MCCAN. The MCCAN predicts a sequence of actions for each agent based on its observation and the meta-command sampled from the Top-$k$ Softmax distribution of CEN. The observations are processed by a deep LSTM, which maintains memory among steps. We use the target attention mechanism to improve the model prediction accuracy, and we design the action mask module to eliminate unnecessary actions for efficient exploration. To manage the uncertain value of state-action in the game, we introduce the multi-head value estimation (Ye et al., 2020a) into the MCCAN by grouping the extrinsic rewards in Table 4. We also treat the intrinsic rewards as a value head and estimate it using a separate value network. Besides, we introduce a value mixer module (Rashid et al., 2018) to model team value to improve the accuracy of the value estimation. All value networks consist of an FC layer with LSTM outputs as input and output a single value, respectively. Finally, following Ye et al. (2020a) and Gao et al. (2021), we adopt hierarchical heads of actions, including three parts: 1) What action to take; 2) who to target; 3) how to act.

**Network Parameter Details.** We develop 6 channels of spatial features read from the game engine with resolution 6*17*17, and then use 5*5 and 3*3 CNN to sequentially extract features. The LSTM unit sizes are 1024 and the time steps are 16. The $k$ value in the CEN sampling is set to 20.

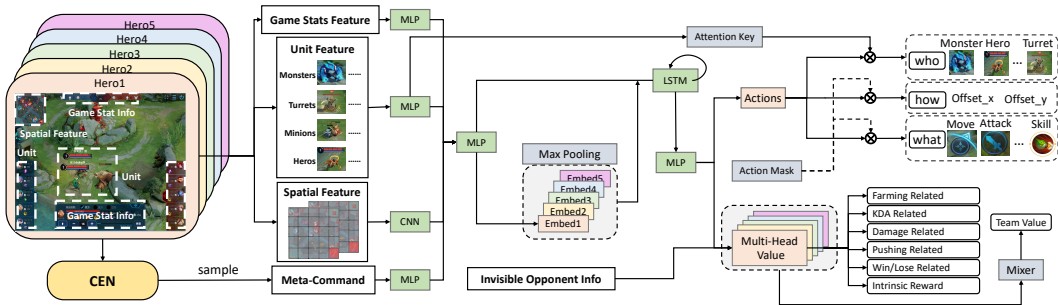

Figure 12: The MCCAN model structure.

### B.5.3 CS

Figure 13 shows the detailed model structure of CS. For each agent, the CS predicts the value $Q(o, C, m)$ or probability $\pi(m|o, C)$ of each meta-command $m$ based on its observations $o$ and all received meta-commands $C$. First, all received meta-commands ($m^H$ from human and $m$ from

the CEN) is reshaped to a 2D image (12*12*1). Then, we use a shared CNN to extract the region-related information from meta-commands, i.e., $z_m^H$ =CNN($m^H$) and $z_m$ =CNN($m$). Besides, the map embeddings of all received meta-commands are integrated into a map set embedding by max-pooling, i.e., $z_{ms}$ =Max-pooling($z_m^H, z_m$). After that, we use the Gating Mechanism to fuse the map set embedding and the state embedding $z_o$ =MLP($o$) of the observation information $o$. For the Gating Mechanism, we use the state embedding $z_o$ and map set embedding $z_{ms}$ to calculate the attention to each other, that is, the gating $g_{ms}$ =MLP($z_o$) for $z_{ms}$ and the gating $g_o$ =MLP($z_{ms}$) for $z_o$. Gating will be used to retain attentional information, as well as fused them, i.e. $z_f = concat[z_o \cdot g_o + z_o; z_{ms} \cdot g_{ms} + z_{ms}]$. Finally, the fused embedding $z_f$ and map embeds $\{z_m^H, z_m\}$ will be used as key $k_f$ and query $\{q_m^H, q_m\}$, respectively, and input into the Target Attention module to predict the state-action value (Q-value) or the probability. We also use the fused embedding to estimate the state value (V-value) from observations $o$ and all meta-commands $C$.

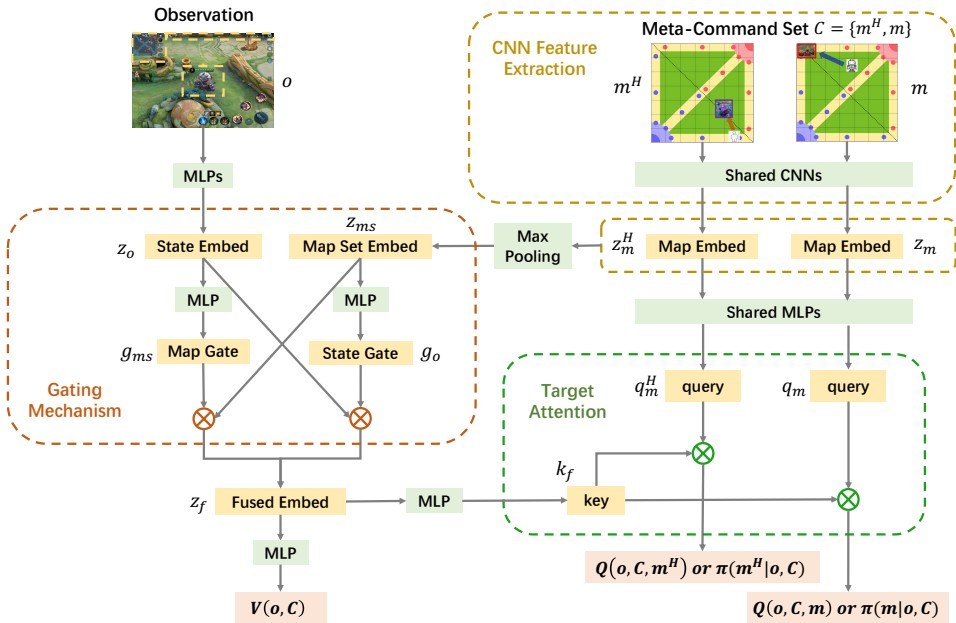

Figure 13: The CS model structure.

## B.6 DETAILS AND ANALYSIS OF COMPONENTS

### B.6.1 META-COMMAND

**Command Converter Function** $f^{cc}$. The $f^{cc}$ extracts the corresponding location and event information from human messages. Specifically, the game engine converts rough messages from the in-game signaling system into predefined protocol buffers (PB), and then the MCC framework uses the $f^{cc}$ to extract the location and event information from the received PB to construct the corresponding meta-command.

**Command Extraction Function** $f^{ce}$. The $f^{ce}$ directly extracts the agent's location from the current state (the position feature in Table 6), which is used to obtain the intrinsic rewards by calculating the "distance" from the agent to the currently executed meta-command.

### B.6.2 CEN

**Training Data**. We extract meta-commands from expert game replay authorized by the game provider, which consist of high-level (top 1% player) license game data without identity information. The input features of CEN are shown in Table 5. The game replay consists of multiple frames, and the information of each frame is shown in Figure 8. We divide the location $L$ of meta-commands in the map into 144 grids. And we set the event $E$ to be all units (103 in total) in the game. For setting $T^{mc}$, we counted the player's completion time for meta-commands from expert game replay, and the

results are shown in Figure. 14. We can see that 80% meta-commands can be completed within the time of 20 seconds in *Honor of Kings*. Thus, $T^{mc}$ is set to 300 time steps (20 seconds).

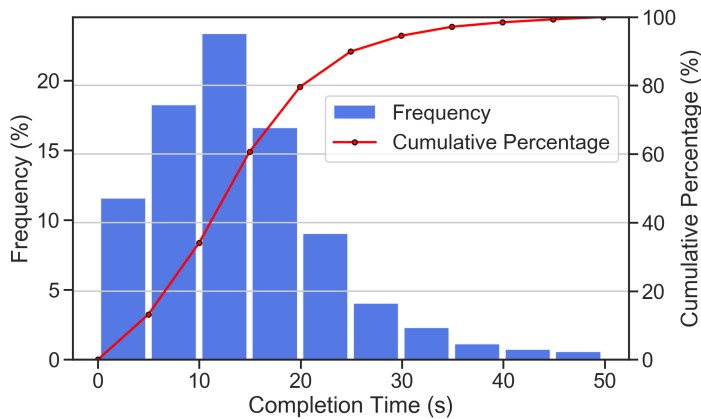

Figure 14: Time statistics of humans completing meta-commands in real games.

Given a state $s_t$ in the trajectory, we first extract the observation $o_t$ for each hero. Then, we use a hand-crafted command extraction function $f^{ce}$ to extract the meta-command $m_t = f^{ce}(s_{t+T^{mc}})$ corresponding to the future state $s_{t+T^{mc}}$. By setting up labels in this way, we expect the CEN $\pi_\phi(m|o)$ to learn the mapping from the observation $o_t$ to its corresponding meta-command $m_t$.

The detailed training data extraction process is as follows:

- First, extract the trajectory $\tau = (s_0, \ldots, s_t, \ldots, s_{t+T^{mc}}, \ldots, s_N)$ from the game replay, where $N$ is the total number of frames.
- Second, randomly sample some frames $\{t | t \in \{0, 1, \ldots, N\}\}$ from the trajectory $\tau$.
- Third, extract feature $o_t$ from state $s_t$ for each frame $\{t | t \in \{0, 1, \ldots, N\}\}$.
- Fourth, extract the location $L_t$ and the event $E_t$ from the state $s_{t+T^{mc}}$ in frame $t + T^{mc}$.
- Fifth, use $L_t$ and $E_t$ to construct the label $m_t = <L_t, E_t, T^{mc}>$.
- Finally, $<o_t, m_t>$ is formed into a pair as a sample in the training data.

**Optimization Objective**. After obtaining the dataset $\{<o, m>\}$, we train the CEN $\pi_\phi(m|o)$ via supervised learning (SL). Due to the imbalance of samples at different locations of the meta-commands, we use the focal loss (Lin et al., 2017) to alleviate this problem. Thus, the optimization objective is:

$$L^{SL}(\phi) = \mathbb{E}_{O,M} \left[ -\alpha m (1 - \pi_\phi(o))^\gamma \log(\pi_\phi(o)) - (1-\alpha)(1-m)\pi_\phi(o)^\gamma \log(1-\pi_\phi(o)) \right],$$

where $\alpha = 0.75$ is the balanced weighting factor for positive class ($m = 1$) and $\gamma = 2$ is the tunable focusing parameter. Adam (Kingma & Ba, 2014) is adopted as the optimizer with an initial learning rate of 0.0001. Especially, We compute the focal loss for $L$ and $E$, respectively.

**Experimental Results**. Figure 15 shows the meta-command distributions of the initial CEN, the converged CEN, and high-level humans. We see that the meta-commands predicted by the CEN gradually converge from chaos to the meta-commands with important positions. The Kullback-Leibler (KL) divergence of the meta-command distribution between the CEN and high-level humans decreases from 4.96 to 0.44 as training converges. The distribution of the converged CEN in Figure 15(b) is close to the distribution of high-level humans in Figure 15(c), suggesting that the CEN can simulate the generation of human meta-commands in real games.

### B.6.3 MCCAN

**Training Environment.** We use a similar training environment as the WuKong agent and the OpenAI Five agent, i.e., the agent interacts with the environment in a self-play manner. Specifically, for

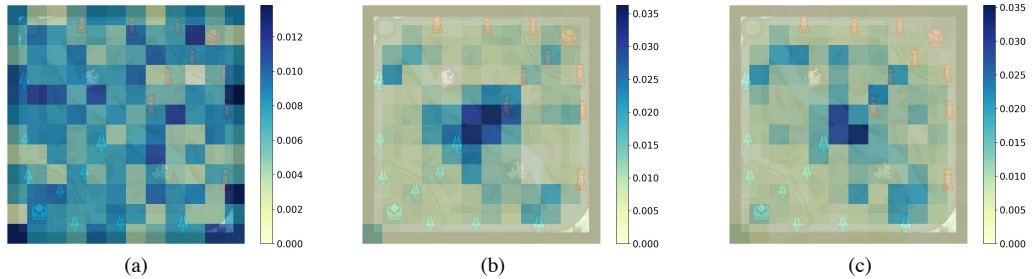

Figure 15: **The meta-command distributions of CEN and high-level humans.** (a) The meta-command distribution of the initial CEN. (b) The meta-command distribution of the converged CEN. (c) The meta-command distribution of high-level humans.

the MCCAN training, each of the 10 agents always executes its own meta-command generated by the CEN. And for each agent, every $T^{mc}$ time steps, the pre-trained CEN $\pi_\phi(m|o)$ generates a meta-command $m_t$ based on its observation $o_t$. The meta-command $m_t$ is then kept for $T^{mc}$ time steps and sent to the MCCAN continuously. During the interval $[t, t + T^{mc}]$, the MCCAN predicts a sequence of actions based on $o_t$ and $m_t$ for the agent to perform.

**Optimization Objective.** The MCCAN is trained with the goal of achieving a higher completion rate for the meta-commands generated by the pre-trained CEN while ensuring that the win rate is not reduced. To achieve this, we introduce extrinsic rewards (including individual and team rewards) and intrinsic rewards

$$r_t^{int}(s_t, m_t, s_{t+1}) = \left|f^{ce}(s_t) - m_t\right| - \left|f^{ce}(s_{t+1}) - m_t\right|,$$

where $f^{ce}$ extracts the agent's location from state $s_t$, and $\left|f^{ce}(s_t) - m_t\right|$ is the distance between the agent's location and the meta-command's location in time step $t$. Intuitively, the intrinsic rewards are adopted to guide the agent to the location $L$ of the meta-command and stay at $L$ to do some event $E$. The extrinsic rewards are adopted to guide the agent to perform optimal actions to reach $L$ and do the optimal event at $L$. Overall, the optimization objective is maximizing the expectation over extrinsic and intrinsic discounted total rewards $G_t = \mathbb{E}_{s \sim d_{\pi_\theta}, a \sim \pi_\theta}\left[\sum_{i=0}^{\infty} \gamma^i r_{t+i} + \alpha \sum_{j=0}^{T^{mc}} \gamma^j r_{t+j}^{int}\right]$, where $d_\pi(s) = \lim_{t \to \infty} P\left(s_t = s \mid s_0, \pi\right)$ is the probability when following $\pi$ for $t$ steps from $s_0$. We use $\alpha$ to weigh the intrinsic rewards and the extrinsic rewards. The experimental results about the influence of $\alpha$ on the performance of the MCCAN are shown in Figure 16.

**Training Process.** The MCCAN is trained by fine-tuning a pre-trained WuKong model (Ye et al., 2020a) conditioned on the meta-command sampled from the pre-trained CEN. Note that the WuKong model is the State-Of-The-Art (SOTA) model in *Honor of Kings*, which can easily beat the high-level human players [1] [2].

We also modified the Dual-clip PPO algorithm (Ye et al., 2020a) to introduce the meta-command $m$ into the policy $\pi_\theta(a_t|o_t, m_t)$ and the advantage estimation $A_t = A(a_t, o_t, m_t)$. The Dual-clip PPO algorithm introduces another clipping parameter $c$ to construct a lower bound for $r_t(\theta) = \frac{\pi_\theta(a_t|o_t, m_t)}{\pi_{\theta_{old}}(a_t|o_t, m_t)}$ when $A_t < 0$ and $r_t(\theta) \gg 0$. Thus, the policy loss is:

$$L^\pi(\theta) = \mathbb{E}_{s,m,a}[\max(cA_t, \min(\text{clip}(r_t(\theta), 1 - \tau, 1 + \tau)A_t, r_t(\theta)A_t)],$$

where $\tau$ is the original clip parameter in PPO. And the multi-head value loss is:

$$L^V(\theta) = \mathbb{E}_{s,m}[\sum_{head_k} (G_t^k - V_\theta^k(o_t, m_t))], V_{total} = \sum_{head_k} w_k V_\theta^k(o_t, m_t),$$

where $w_k$ is the weight of the $k$-th head and $V_t^k(o_t, m_t)$ is the $k$-th value.

---

[1]Wukong AI beats human players to win Honour Of Kings mobile game, https://www.thestar.com.my/tech/tech-news/2019/08/07/tencent039s-ai-beats-human-players-to-win-honour-of-kings-mobile-game

[2]Honor of Kings Wukong AI 3:1 defeated the human team, https://www.sportsbusinessjournal.com/Esports/Sections/Technology/2021/07/HoK-AI-Battle.aspx?hl=KPL&sc=0

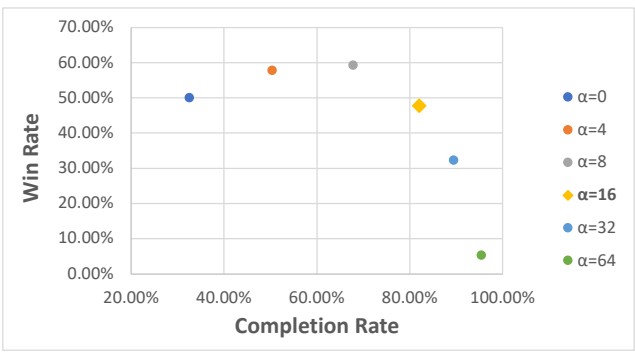

Figure 16: The win rate and completion rate of MCCAN with different $\alpha$. The opponent is the pre-trained action network, i.e., MCCAN with $\alpha$=0.

**Experimental Results**. We conducted experiments to explore the influence of the extrinsic and intrinsic reward trade-off parameter $\alpha$ on the performance of MCCAN. The win rate and completion rate results are shown in Figure 16. As $\alpha$ increases, the completion rate of MCCAN gradually increases, and the win rate of MCCAN first increases and then decreases rapidly. Notably, we can train an agent with a completion rate close to 100% by increasing $\alpha$, but this will significantly reduce the win rate because the meta-command executed is not necessarily optimal and may result in the death of agents.

When $\alpha = 16$, the completion rate of the trained agent for meta-commands is 82%, which is close to the completion rate of humans (80%). And the win rate of the trained agent against the SOTA agents (Ye et al., 2020a; Gao et al., 2021) is close to 50%. Thus, we finally set $\alpha = 16$ .

### B.6.4 CS

**Additional Results in Agent-Only Collaboration Testing.** To eliminate the transitivity of WR, we added additional experiments to compare the MCC agent and baseline agents to the WuKong (SOTA) agent for each 600 matches in Test I Environment. The WRs (P1 versus P2) are shown in Table 9. We see that the effective CS mechanism in the MCC agent enables the agents to communicate and collaborate effectively, which improves the WR compared to the WuKong agent (SOTA).

Table 9: The WRs of the MCC agent and baseline agents against the WuKong agent (P1 versus P2).

| P2\P1 | MC-Base | MC-Rand | MC-Rule | MCC |
|---|---|---|---|---|
| WuKong | 49.2% | 18.5% | 39.5% | 57.8% |

**Ablation Studies.** We further investigate the influence of different components on the performance of CS, including CNN feature extraction with the gating mechanism (w/o CNN-GM), target attention module (w/o TA), and PPO optimization algorithm (MCC-PPO). We conduct ablation studies in Test I with a 20 hero pool. In practical games, meta-commands with adjacent regions often have similar intentions and values. Thus the response rate of the agent to adjacent meta-commands should be as close as possible. Besides, the higher the agent's response rate to meta-commands, the more collaborative the agent's behaviors. Thus we expect the response rate of CS to be as high as possible. Generally, we expect CS's Response Rate (RR) to be as high as possible while ensuring that the Win Rate (WR) maintains.

Figure 17(a) demonstrates the WR of different CS ablation versions during the training process, and Figure 17(b) shows the converged WR-RR results. We see that after ablating the TA module, the WR and RR of CS reduce significantly, indicating that the TA module can improve the accuracy of CS to meta-commands. Besides, after ablating the CNN-GM module, the RR of CS is most affected, which is reduced by 20%. It indicates that without the CNN-GM module, the value estimation of CS to adjacent meta-commands is not accurate enough, resulting in missing some highly valuable meta-commands. We notice that the MCC and MCC-PPO in both metrics are close, confirming the versatility of the CS model structure.

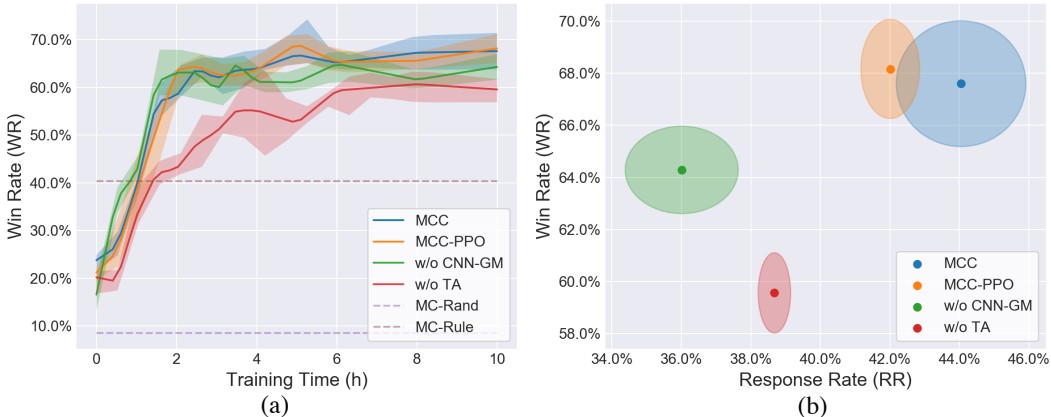

Figure 17: **Results of ablation studies.** (a) The training curves of different CS ablation versions. (b) The converged WR-RR results of different CS ablation versions. The shadow indicates the standard deviation.

## C  DETAILS OF HUMAN-AGENT COLLABORATION TEST

### C.1  ETHICAL REVIEW

The ethics committee of a third-party organization, Tencent (the parent company of *Honor of Kings*), conducted an ethical review of our project. They reviewed our experimental procedures and risk avoidance methods (see Appendix C.1.2). They believed that our project complies with the "New Generation of AI Ethics Code" [3] of the country to which the participants belonged (China), so they approved our study. In addition, all participants consented to the experiment and provided informed consent (see Appendix C.1.1) for the study.

#### C.1.1  INFORMED CONSENT

All participants were told the following experiment guidelines before testing:

- This experiment is to study human-agent collaboration technology in MOBA games.
- Your identity information will not be disclosed to anyone.
- All game statistics are only used for academic research.
- You will be invited into matches where your opponents and teammates are agents.
- Your goal is to win the game as much as possible by collaborating with agent teammates.
- You can communicate and collaborate with agent teammates through the in-game signaling system.
- Agent teammates will also send you signals representing their macro-strategies, and you can judge whether to execute them based on your value system.
- After each test, you can report your preference over the agent teammates.
- Each game lasts 10-20 minutes.
- You may voluntarily choose whether to take the test. You can terminate the test at any time if you feel unwell during the test.
- After all tests are complete, you may voluntarily fill out a debrief questionnaire to tell us your open-ended feedback on the experiment.
- At any time, if you want to delete your data, you can contact the game provider directly to delete it.

If participants volunteer to take the test, they will first provide written informed consent, then we will provide them with the equipment and game account, and the test will begin.

---

[3]China: MOST issues New Generation of AI Ethics Code, `https://www.dataguidance.com/news/china-most-issues-new-generation-ai-ethics-code`

### C.1.2 POTENTIAL PARTICIPANT RISKS

First, we analyze the risks of this experiment to the participants. The potential participant risks of the experiment mainly include the leakage of identity information and the time cost. And we have taken a series of measures to minimize these risks.

**Identity Information.** A series of measures have been taken to avoid this risk:

- All participants will be recruited with the help of a third party (the game provider of *Honor of Kings*), and we do not have access to participants' identities.
- We make a risk statement for participants and sign an identity information confidentiality agreement under the supervision of a third party.
- We only use unidentifiable game statistics in our research, which are obtained from third parties.
- Special equipment and game accounts are provided to the participants to prevent equipment and account information leakage.
- The identity information of all participants is not disclosed to the public.

**Time Cost.** We will pay participants to compensate for their time costs. Participants receive $5 at the end of each game test, and the winner will receive an additional $2. Each game test takes approximately 10 to 20 minutes, and participants can get about an average of $20 an hour.

## C.2 EXPERIMENTAL DETAILS

### C.2.1 PARTICIPANT DETAILS

We contacted the game provider and got a test authorization. The game provider helped us recruit 30 experienced participants with personal information stripped, including 15 high-level (top1%) and 15 general-level (top30%) participants. All participants have more than three years of experience in *Honor of Kings* and promise to be familiar with all mechanics in the game, including the in-game signaling system in Figure 9.

And special equipment and game accounts are provided to each participant to prevent equipment and account information leakage. The game statistics we collect are only for experimental purposes and are not disclosed to the public.

### C.2.2 EXPERIMENTAL DESIGN

We used a within-participant design: *m Human + n Agent* (*mH + nA*) team mode to evaluate the performance of agents teaming up with different numbers of participants, where $m + n = 5$. Each participant is asked to randomly team up with three different types of agents, including the MC-Base agents, the MC-Rand agents, and the MCC agents. For fair comparisons, participants were not told the type of their agent teammates. The MC-Base agent team was adopted as the fixed opponent for all tests. Participants tested 20 matches for the *1H + 4A* team mode. High-level participants tested additional 10 matches for the *2H + 3A* and the *3H + 2A* team modes, respectively. After each game test, participants reported their preference over the agent teammates.

We prohibit communication between agents to eliminate the effects of collaboration. Thus the agents can only communicate with their human teammates. In each game test, humans can send the converted meta-commands whenever they think their macro-strategies are important. Furthermore, to make the agent behave like humans (at most one human sends his/her meta-command at a time step), we restricted agents from sending their meta-commands, i.e., only one agent sends a valuable meta-command to human teammates from the agents' current meta-commands. Specifically, this process consists of several steps: (1) The MCC framework randomly chooses a human teammate (note that human and agent observations are shared); (2) The MCC framework uses his/her observation and all agents' meta-commands as the input of the CS, and obtained the estimated values of these meta-commands; (3) The MCC framework selects the meta-command with the highest value; (4) The MCC framework selects the corresponding agent to send the meta-command. The above process is executed at an interval of 20 seconds.

In addition, as mentioned in Ye et al. (2020a); Gao et al. (2021), the response time of agents is usually set to 193ms, including observation delay (133ms) and response delay (60ms). The average APM of

agents and top e-sport players are usually comparable (80.5 and 80.3, respectively). To make our test results more accurate, we adjusted the agents' capability to match high-level humans' performance by increasing the observation delay (from 133ms to 200ms) and response delay (from 60ms to 120 ms).

### C.2.3 PREFERENCE DESCRIPTION

After each test, participants gave scores on several subjective preference metrics to evaluate their agent teammates, including the **Reasonableness of H2A** (how well agents respond to the meta-commands sent by participants), the **Reasonableness of A2H** (how reasonable the meta-commands sent by agents are), and the **Overall Preference** for agent teammates.

For each metric, we provide a detailed problem description and a description of the reference scale for the score. Participants rated their agent teammates based on how well their subjective feelings matched the descriptions in the test. The different metrics are described as follows:

- For the Reasonableness of H2A, "Do you think the agent teammates respond reasonably to your commands? Please evaluate the reasonableness according to the following scales."
    1) Terrible: No response, or totally unreasonable.
    2) Poor: Little response, or mostly unreasonable.
    3) Normal: Response, but some unreasonable.
    4) Good: Response, mostly reasonable.
    5) Perfect: Response, and perfect.

- For the Reasonableness of A2H, "Do you think the commands sent by the agent teammates are reasonable? Please evaluate the reasonableness according to the following scales. Note that if you don't receive any commands, please ignore this question."
    1) Terrible: Totally unreasonable.
    2) Poor: Low reasonable.
    3) Normal: Some reasonable.
    4) Good: Mostly reasonable.
    5) Perfect: Totally reasonable.

- For the Overall Preference, "What is your overall preference for the agent teammates collaborating with you? Please rate the following words according to your subjective feelings: 1) Terrible; 2) Poor; 3) Normal; 4) Good; 5) Perfect".

### C.2.4 ADDITIONAL SUBJECTIVE PREFERENCE RESULTS

Detailed subjective preference statistics are presented in Table 10. Compared with no collaboration (MC-Base) and random collaboration (MC-Rand), the participants preferred reasonable collaboration (MCC).

**Reasonableness of H2A.** Participants prefer MC-Rand over MC-Base, suggesting that they expect the agent to respond more to their commands. Nonetheless, the score of MCC is much higher than MC-Rand, indicating that participants prefer their commands to get reasonable rather than incorrect responses.

**Reasonableness of A2H.** Participants express a significant preference for MCC over MC-Rand, demonstrating that they agree with MCC's commands and are more willing to collaborate. The results are consistent with Figure 7, where participants believe the commands sent from MCC align more with their own value system. Note that the non-collaborative setting prohibits MC-Base from sending any commands, so we made no statistics.

**Overall Preference.** Participants are satisfied with the MCC agent over other agents and give the highest score. By comparing MC-Base and MCC, we can observe that human-agent collaboration based on meta-command communication can bring a better impression of the human-agent team in the MOBA game. However, while MC-Rand is higher than MC-Base in the Reasonableness of H2A metric, it is lower than MC-Base in the Overall Preference metric. Therefore, collaboration is important but not as necessary as winning. This metric further confirms the reasonableness of MCC in human-agent collaboration.

Table 10: The subjective preference results of all participants in the Human-Agent Collaboration Test, including mean scores and variances (in parentheses).

| Participant Preference Metrics (from terrible to perfect, 1∼5) | Human Teammate | Type of Agent | | |
|---|---|---|---|---|
| | | MC-Base | MC-Rand | MCC |
| Reasonableness of H2A | General-level | 2.3 (0.38) | 2.7 (0.24) | **4.0** (0.60) |
| | High-level | 2.2 (0.21) | 2.5 (0.41) | **4.1** (0.55) |
| Reasonableness of A2H | General-level | - | 1.9 (0.35) | **4.3** (0.31) |
| | High-level | - | 1.7 (0.24) | **4.4** (0.35) |
| Overall Preference | General-level | 2.7 (0.41) | 1.3 (0.27) | **4.3** (0.40) |
| | High-level | 2.5 (0.21) | 1.2 (0.17) | **4.5** (0.41) |

# D  DISCUSSION

**Limitations and future work.** First, the training process of the MCC agent consumes vast computing resources like other SOTA MOBA agents. Thus, we will optimize the training process of existing MOBA agents, aiming to lower the threshold for researchers to study and reproduce work in MOBA games. Second, the meta-commands we proposed are generic to MOBA games and cannot be directly extended to other types of games, such as FPS and Massively Multiplayer Online (MMO). We will design a more general meta-command representation, such as natural language, and extend the MCC framework to other games. Third, we will apply the MCC agents to the friendly bots in the teaching mode of *Honor of Kings*. All in all, we would hope that this work can not only offer new ideas to researchers but also bring a better human-agent collaboration experience in practical applications.

