# OpenReview forum: "Towards Effective and Interpretable Human-Agent Collaboration in MOBA Games: A Communication Perspective"
_ICLR.cc/2023/Conference — ICLR 2023 notable top 25%_

### Official Review · Reviewer_Usav · 2022-10-24

**Confidence:** 4
**Correctness:** 3
**Technical Novelty And Significance:** 3
**Empirical Novelty And Significance:** 3
**Recommendation:** 8

**Clarity, Quality, Novelty And Reproducibility:**

### Quality and technical details
- The baselines used for the experiments are rather poor:
1. In order to put the win rates in context, what are the win rates for 5A teams?
2. MC Rand seems useful as a sanity check only, and MC Base agents do not communicate at all. Hence, the relative results in the subjective experiments are not very illustrative. Preference of collaborating with other human (general and high) players is required, in other words, do human players prefer MCC over other human players?
- Shouldn't $G_t$ include the expectation over meta-commands?
- Why is the message $m_t^{i+1}$ dependent only on the observation, $o_t^i$, and not on the selected message, $c_t^i$?
- It is said "MC-Base agent has the same capabilities as the WuKong agent". However, the MC Base agents have been fine-tuned, could fine tuning have decreased the performance w.r.t. the original WuKong agent?

## Missing reproducibility details
- Please include details on the hand-crafted command converter function $f^{cc}$ and the hand-crafted command extraction function $f^{ce}$. Could you please be more explicit how these functions relate to Tables 4, 5 and 6?
- Tables 5, 6 and 7 describe the data type and the dimensionality, could you be more specific on the values (for example, scalar value refers to the number of elements, is it a normalised value, etc.)?
- Please provide details on the value networks $V_t^{int}(o_t, m_t)$ and $V_t^{ext}(o_t)$ used to train the meta-command conditioned action network.
- Appendix B.5 explains high level details of the networks, but lower-level details are missing, like the number of hidden layers, their size and activation functions, the value of $k$ for the Top-$k$ SoftMax activation. In addition, explanations on the rationale for choosing these parameters would be much appreciated.
- Please provide details on the model used for cloning the expert data (Sec. B.6.1 describes the loss function but not the model). Also, what is the total number of frames, $N$, per trajectory?
- Which RL algorithm is used to train the Command Selector network? (PPO is mentioned for the low-level actions (MCCAN) but haven't found a similar discussion for the CS network).
- Training details are also missing, for example B.6.1 mentions the initial learning rate, but says nothing about the learning rate schedule or the other parameters.
- According to Figure 3, $V_t^{mc}$ and $\{Q_t^{i}\}$ seem to refer to multiple heads. Does $V_t^{mc}$ propagate to the shared CNN? Do the h$\{Q_t^{i}\}$ heads propagate to the gating and observation MLPs?
- In Figure 4(b), Test II, there is a simulated human. How has this agent been trained
- Is the number of steps at the human-agent collaboration stage, $n$, fixed? or is it dependent on the meta-command? Can a command be interrupted by another meta-command (e.g., when an agent executing "kill the dragon" that could take longer than 20 seconds receives other messages every 20 secs)?
- How are the resources in each cluster used? Are the 63000 CPUs used to, e.g., run about 2.3 episodes each to feed the 560 GPUs?


### Clarity
- Notation is sometimes inconsistent. For example, is $V(h) = V_t^{mc}(o_t, C_t)$? Do we have one state-action value network per human and agent sending messages, right? Does $Q(h, m')$ refer to each $Q_t^{i}(o_t, C_t, m_t^{i})$ for $i=1\ldots M+N$ or to a single loss after combining the different attention heads?
- Over which distribution is taken the expected value in the definition of the value of the meta-command, $L^V$? In other words what are $S$ and $C$ in $V_\omega^k(S,C)$ I understand $C$ is the meta-command candidate set, is $S$ the current state?



### Minor comments
- What does “physical” in a "physical computer cluster" mean?
- Could you give some insight on how the gating mechanism works for completeness? Does the attention mechanism establish which meta-commands are more relevant for the current observation?





**Details Of Ethics Concerns:**

The authors introduce a simple methodology with excellent results, both in terms of winning rate and subjective preference, and I am concerned this might potentially be used to collaborate with drones and other AI agents in real life combat scenarios, and wonder whether this should be reviewed by the Ethics Committee.


**Strength And Weaknesses:**

### Strengths
+ The proposed framework is conceptually simple and might be extended to other scenarios (beyond MOBA or even beyond gaming) where humans and agents communicate at the strategic level.
+ The whole design is full of interesting design choices that will likely inspire future research, like the reward function to fine tuning the low-level actions network, the command representation as a grid that it can be learnt with a CNN, or the command selector architecture that combines the gating mechanism with attention
+ The paper establishes a baseline for a challenging and relevant problem.

### Weaknesses
- Many implementation details are missing (see comments on reproducibility below).
- Clarity could be improved (see comments below)

**Summary Of The Paper:**

The paper proposes a framework for human-agent collaboration in MOBA games. The framework consists in two main parts: i) a protocol of commands that human and agents use to communicate with each other, and that can be expressed as a tuple of "what" the sender asks the receiver to do, for "how long" has to be done, and "where" the action has to happen; and ii) a hierarchical RL approach the agents follow to learn what commands to send, and to which commands they should respond - the approach is hierarchical because the learning process is done sequentially in three independent stages: first, message generation is leant by cloning the behaviour of expert data; second, the low level actions required to accomplish a command are learnt by fine tuning a pretrained model, such that the agents learn to complete the tasks described by the commands without decreasing the win rate; third, the agents learn to select which message among all the messages received (e.g., one per human player). Experiments show the proposed framework results in increased win rates with respect to agents that do not communicate with other players (i.e., they only execute their own commands), and that human players prefer the proposed agents with respect to agents that do not communicate.


**Summary Of The Review:**

**Edit after the authors rebuttal**

The authors have addressed my concerns as discussed in the rebuttal thread. Although the proposed approach is MOBA specific, I think it might inspire extensions to other game and real scenarios - see my ethical concern below. Moreover, the results in terms of winning rate and subjective preference are impressive. I am raising my score to acceptance. However, as noted in the rebuttal thread, I agree with other reviewers there are some issues to be discussed.

**Original review**

This is an interesting paper that tackles an important problem in a rather sophisticated manner. It has good ideas and can likely inspire future research.

The results are illustrative but not strong. Fortunately, win rates of 5A teams and preference over playing with other human players are strong baselines that would shed light on the performance of the proposed solution.

In my opinion, a major contribution of this work is to actually make this thing work, which includes multiple machine learning engineering decisions and engineering expertise. However, although some high-level ideas are given, many crucial details are missing, which could prevent reproducing the results, and hence reduce the potential impact of the work.

If the authors tackle these two concerns: stronger baselines and reproducibility, I will increase my recommendation score.

---

> ### Author Response · Authors · 2022-11-14
> **Response to Reviewer Usav (1/2)**
>
> Thank you for carefully reviewing our paper! We provide clarification below for your questions and concerns. If you have any further questions or comments, we will be happy to discuss them further.
>
> ---
>
> **Clarify on the baselines**
>
> Please let me clarify the WuKong and MC-Base agents in detail.
>
> - The WuKong agent [1]: The SOTA agent in HoK, which can easily beat the high-level human players [2,3]. Before our Human-Agent Collaboration Test, we conducted experiments on 5A (WuKong) teams against 5H (high-level) teams across 50 matches and obtained a **winning rate of 98% (5A teams versus 5H teams)**.
>
> - The MC-Base agent: It can be considered as CEN + MCCAN ($\alpha=16$), which is finetuned from the WuKong model conditioned on the meta-command sampled from the CEN. Note that the MCCAN ($\alpha=0$) is actually the WuKong model, which has not been finetuned. As Figure 16 shows, **the winning rate of the MC-Base agent against the WuKong agent (MCCAN $\alpha=0$) is close to 50%**. Thus, the MC-Base agent maintains the same capabilities as the WuKong agent.
>
> **Participants prefer MCC agents even more than themselves**.
>
> - Table 2 shows that the RRs of both high-level and general-level participants (Receiver) to MCC agents (Sender) are 78.5% and 73.43%, respectively, close to and even higher than those of participants themselves (74.91% and 72.34%), indicating that participants are willing to respond to the MCC agents' meta-commands and collaborate with them.
>
> - The Overall Preference results in Figure 6 also demonstrate that the participants prefer to play with the MCC agent than others.
> Besides, we collected open-ended feedback on their experience from participants. Some high-level participants commented on the MCC agents, "They can understand my intentions well and cooperate with me tacitly. This is an amazing and unforgettable experience". Some general-level participants commented, "They are not just skilled teammates, they are more of a coach. I was able to learn a lot of strategies that can help me win."
>
> ---
>
> **Reproducibility details**
>
> **Q1**: Provide details on the command converter function $f^{cc}$ and the command extraction function $f^{ce}$.
>
> **A1**: The $f^{cc}$ extracts the corresponding location and event information from human messages. Specifically, the game engine converts rough messages from the in-game signaling system into predefined protocol buffers (PB), and then the MCC framework uses the $f^{cc}$ to extract the location and event information from the received PB to construct the corresponding meta-command.  The $f^{ce}$ directly extracts the agent's location from the current state (the position feature in Table 6), which is used to obtain the intrinsic rewards by calculating the "distance" from the agent to the currently executed meta-command. We have supplemented the descriptions of $f^{cc}$ and $f^{ce}$ in Appendix B.6.1.
>
> ---
>
> **Q2**: Provide more specific on the values in Tables 5, 6, and 7.
>
> **A2**: We have added a description of the data value in Tables 5, 6, and 7, and marked it in red. Please see our rebuttal revision.
>
> ---
>
> **Q3**: Provide details on the value networks $V_t^{int}(o_t, m_t)$ and $V_t^{ext}(o_t, m_t)$ of the MCCAN.
>
> **A3**: As shown in Figure 12, to manage the uncertain value of state-action in the game, we introduce the multi-head value estimation [1] into the MCCAN by grouping the extrinsic rewards in Table 4. We also treat the intrinsic rewards as a value head and estimate it using a separate value network. All value networks (including $V_t^{ext}(o_t, m_t)$ and $V_t^{int}(o_t, m_t)$) consist of an FC layer with LSTM outputs as input (1024 dimensions) and output a single value, respectively. We have supplemented these descriptions in Appendix B.5.2.
>
> ---
>
> **Q4**: Provide details on the CEN, the MCCAN, the CS, and parameter settings.
>
> **A4**: We have supplemented the relevant details in Appendix B.5 and released the MCC framework code including the CEN network, the MCCAN network, the CS network, and the detailed parameter settings. Please see at https://github.com/ICLR2023-MCC/MCC.
>
> ---
>
> **Q5**: Provide details on the description of the CEN model. What is the total number of frames $N$ per trajectory?
>
> **A5**: We have added the additional description of the CEN network in Appendix B.5.1. A trajectory is a whole game in fact. The total number of frames $N$ is the total number of frames per game (not fixed).
>
> ---
>
> **Q6**: Which RL algorithm is used to train the CS?
>
> **A6**: We used two popular RL algorithms, i.e., PPO [4] and DQN [5], to train the CS. In Section 3.3, we mentioned that "the CS model structure can be easily applied to most popular RL algorithms, such as PPO, DQN, etc.". In Appendix B.6.4, we conducted Ablation Studies to investigate the influence of different components including optimization algorithms (PPO and DQN) on the performance of CS.

---

> > ### Author Response · Authors · 2022-11-14
> > **Response to Reviewer Usav (2/2)**
> >
> > **Q7**: Does $V_t^{mc}$ propagate to the shared CNN? Do the $Q_t^i$ heads propagate to the gating and observation MLPs ?
> >
> > **A7**: To clearly illustrate the network structure and working of CS, we have added a detailed description of CS in Appendix B.5.3. The total loss of CS consists of V loss and Q loss, all of which update each layer of the network including the shared CNN, the gating, and the observation MLPs through backpropagation.
> >
> > ---
> >
> > **Q8**: In Figure4(b), Test II, how has this agent been trained?
> >
> > **A8**: Test II is a simple test environment that simulates most practical game scenarios, i.e., at most one human can send a meta-command at a time step. The "simulated human" refers to the MCC agent that plays the "human" role of sending meta-commands and not receiving meta-commands from others.
> >
> > ---
> >
> > **Q9**: Is the number of steps at the human-agent collaboration stage, $n$, fixed? Can a command be interrupted by another meta-command?
> >
> > **A9**: $n$, i.e., $T^{mc}$, refers to how often meta-commands are sent. $T^{mc}$ is set to 20 seconds during the training process of the MCC agent and Agent-Only Collaboration testing. During Human-Agent Collaboration testing, humans can send their meta-commands at any time. If the MCC agent receives new meta-commands during the execution of the currently selected meta-command, the MCC agent will estimate the values of new meta-commands and compare them with the currently executing one. If there is a new meta-command with a higher value, the MCC agent will replace the current meta-command with it and continue to execute. If the MCC agent does not receive new meta-commands after executing the current meta-command, the MCC agent will continue to execute its own meta-command generated by the CEN.
> >
> > ---
> >
> > **Q10**: How are the resources in each cluster used?
> >
> > **A10**: The agents perform self-play and interact with the game engine on the AI Server using CPUs to generate Experience. About half of GPUs are used as the Inference Server to generate actions for agents by centralized batch inference. The Experience containing sequences of observations, actions, meta-commands, rewards, etc., are passed asynchronously from AI Server to Memory Pool, which is implemented as a memory-efficient circular queue. The RL learner is a distributed training environment for training models using the other half of GPUs. Every two minutes, the RL Learner distributes the latest models to the Inference Server.
> >
> > ---
> >
> > **Other issues**
> >
> > **Q1**: Shouldn't $G_t$ include the expectation over meta-commands?
> >
> > **A1**: The MCCAN learns how to execute a selected meta-command, thus the corresponding $G_t$ is the expected cumulative rewards including intrinsic rewards (guide the agent to reach $L$ and stay at $L$) and extrinsic rewards (guide the agent to perform optimal actions to reach $L$ and do optimal event $E$ at $L$) during the meta-command execution. The CS learns to select the optimal meta-command for the MCCAN, thus the corresponding $G_t^{mc}$ is the expected cumulative meta-command execution rewards over meta-commands.
> >
> > ---
> >
> > **Q2**: Why is the message $m_t^{i+1}$ dependent only on the observation, $o_t^i$, and not on the selected message, $c_t^i$?
> >
> > **A2**: $m_{t+1} = \pi_{\phi}(o_t)$ is a relatively simple yet effective representation of the meta-commands, that is, the long-term intentions of the future in the current state. Thus, the CEN training data only includes $o_t$ (extracted from $s_t$) and $m_t$ (extracted from $s_{t+T^{mc}}$). More complex representations of the meta-commands such as $m_{t+1} = \pi_{\phi}(o_t, c_t)$ are also reasonable but may face many difficulties in the process of implementation, e.g., how to extract $c_t$ reasonably. We leave this as future work to explore.
> >
> > ---
> >
> > **Q3**: Clarify on Notation.
> >
> > **A3**: We have revised and unified some notations including $V(h)$, $Q(h,m')$, and $V_{\omega}^k(S,C)$. Please see our rebuttal revision (Section 3.3 and Figure 3).
> >
> > ---
> >
> > **Q4**: How the gating mechanism in the CS module works?
> >
> > **A4**: Fuse Embed = CONCAT [State Embed * FC (MapSet Embed) + State Embed, MapSet Embed * FC (State Embed) + MapSet Embed].
> > The main function of the gating mechanism is to screen for relevant features.
> >
> > ---
> >
> > [1] Deheng Ye, et al. "Towards playing full moba games with deep reinforcement learning". NeurIPS'2020.
> >
> > [2] Wukong AI beats human players to win Honour Of Kings mobile game. https://www.thestar.com.my/tech/tech-news/2019/08/07/tencent039s-ai-beats-human-players-to-win-honour-of-kings-mobile-game
> >
> > [3] Honor of Kings Wukong AI 3:1 defeated the human tehttps://www.sportsbusinessjournal.com/Esports/Sections/Technology/2021/07/HoK-AI-Battle.aspx?hl=KPL&sc=0c=0
> >
> > [4] John Schulman, et al. "Proximal policy optimization algorithms". arXiv'2017.
> >
> > [5] Volodymyr Mnih, et al. "Human-level control through deep reinforcement learning". Nature'2015.

---

> > > ### Comment · Reviewer_Usav · 2022-11-20
> > > **Thank for the detailed responses**
> > >
> > > I thank the authors for addressing all my comments. The authors have explained the provided baselines are strong. The authors have also made an effort to make their results reproducible, giving more details in the main text and appendixes, as well as releasing the code; the data and game engine have not been released arguing they are proprietary and confidential, but I consider releasing the code fair enough.
> > >
> > > I think the proposed solution is conceptually simple, and sophisticated from an engineering point of view. Moreover, the results are significant. Hence, I am inclined to raise my recommendation score to accept.
> > >
> > > On the other hand, Reviewer peTd has raised concerns on the clarity with which I agree. I had a hard time understanding the method and some of the details. This might make me reduce my score during the discussion period.
> > >
> > > In addition, due precisely to the excellent results, both in terms of winning rate and preference, I would like to raise an ethical concern, regarding the potential harm that the application of this technology could have if combined with drones and other AI agents in real combat scenarios.

---

> > > > ### Author Response · Authors · 2022-11-21
> > > > **Re: Thanks for your further comments**
> > > >
> > > > We appreciate your positive comments on the MCC framework and the experimental study. Thank you so much for improving your score, and for pointing out the clarity issue and the ethical issue. We provide explanations below for your concerns.
> > > >
> > > > **Clarification**
> > > >
> > > > We thank you and other reviewers for pointing out the lack of clarification on the MCC framework. We have made detailed clarifications on the points including Meta-Command, CEN, MCCAN, and CS currently raised by all the reviewers, and included them in the paper and appendix. If you and other reviewers have new concerns about the paper that need to be clarified further during the discussion period, we would be happy and definitely provide thorough and detailed explanations in the paper and appendix. We believe that after including these clarifications, the revised paper would be more accessible and the quality can be reinforced.
> > > >
> > > > **Ethical Concerns**
> > > >
> > > > We empathize with your concerns about ethical issues. Below we give some explanations, hoping to alleviate your concerns in this regard.
> > > >
> > > > First, this research project has passed the ethical review of a third-party organization (Tencent, the parent company of HoK) and complies with the "Artificial Intelligence Ethics Code" of the country to which the organization belongs. So, our research work was approved to conduct. For more ethical details, please see Appendix C.1.
> > > >
> > > > Second, the proposed MCC framework is currently only generic to MOBA games. Like other research work FTW[1], AlphaStar[2], and OpenAI Five[3], it is almost impossible to directly apply the MCC framework to real combat scenarios, because of many Sim-to-Real problems, such as environment simulator and modeling errors.
> > > >
> > > > Third, from an academic research perspective, we expect that this work can contribute to the AI community like other influential research in game AI, such as FTW[1], AlphaStar[2], and OpenAI Five[3]. And we hope that this work would inspire researchers to enter the research area of human-agent collaboration in complex environments. From an industrial application perspective, we will precipitate this work and apply its research outcome to the friendly bots in the teaching mode of HoK, aiming to provide gameplay teaching to novice players.
> > > >
> > > > ---
> > > >
> > > > [1] Jaderberg, Max, et al. "Human-level performance in 3D multiplayer games with population-based reinforcement learning". Science'2019.
> > > >
> > > > [2] Oriol Vinyals, et al. "Grandmaster level in StarCraft II using multi-agent reinforcment learning". Nature'2019.
> > > >
> > > > [3] OpenAI, et al. "Dota 2 with large scale deep reinforcement learning". arXiv'2019.

---

> > > > > ### Comment · Reviewer_Usav · 2022-11-28
> > > > > **Some clarity issues still remain**
> > > > >
> > > > > I thank the authors for their response. Although I have raised my score, I agree with other reviewers that some important clarity issues remain, which may make me reduce my recommendation score back to 6: borderline accept.
> > > > >
> > > > > For example, the authors refer to the metacommand as a tuple $<L, E, T>$, but based on the description the output of CEN seems to be a soft-max over the $L$ locations on the grid. This is consistent with other parts of the text that say that $E$ is not explicitly communicated or learned, but it is learned as the sequence of microactions based on the agent's value estimates; moreover, $T$ is set to a fixed value that is useful 80% of the time. Is my understanding correct? If so, the authors should remove any ambiguity and clearly specify that what the CEN module outputs the location rather than the whole metacommand.
> > > > >
> > > > > Regarding the ethical concerns, my point is just that it would be good to get an ethics review from experts on the topic.

---

> > > > > > ### Author Response · Authors · 2022-11-29
> > > > > > **Re: Thanks for your further feedback**
> > > > > >
> > > > > > Dear Reviewer Usav,
> > > > > >
> > > > > > We appreciate your quick response to help us continually improve this work. And thanks for pointing out the clarity issue.
> > > > > >
> > > > > > ---
> > > > > >
> > > > > > > Clarity on the CEN module outputs
> > > > > >
> > > > > > **The CEN module receives observation $o$ as input and outputs the distribution of meta-commands $m$ (including $L$ and $E$, $E$ is conditioned on $L$).**
> > > > > >
> > > > > > For training the CEN, we extract the observation $o_t$ from the state $s_t$ and extract the meta-command $m_t$ including $L_t$ and $E_t$ from the state $s_{t+T^{mc}}$ in each trajectory to construct the training dataset. We divide the location $L$ of meta-commands in the map into 144 grids. To make the representations learned by CEN more relevant to the macro-strategy (where to go and what to do), we set an event set to be all attackable units (a subset of event space) in the game and only use the prediction of $E$ as an auxiliary task for the CEN training. In the MCC framework, we train the MCCAN to learn to do optimal event $E^*$ at location $L$. Thus, we use $E^*$ instead of using the $E$ outputted by CEN.
> > > > > >
> > > > > > We have revisited the paper, revised the CEN statements to remove possible ambiguity, and included these descriptions in Section 3.2 and Appendix B.6.2.
> > > > > >
> > > > > > ---
> > > > > >
> > > > > > > Clarity on the Meta-Command Communication
> > > > > >
> > > > > > **At the Meta-Command Communication stage, the whole meta-command $<L,E,T>$ is used to communicate, rather than just $L$.** Please allow us to give a detailed explanation of this.
> > > > > >
> > > > > > Before Communication, the MCC framework converts humans' explicit messages into meta-commands $m^H=<L^H, E^H, T^{mc}>$ and uses the CEN to generate meta-commands $m=<L,E,T^{mc}>$ for agents. Since the macro-strategy space is enormous [1], customizing corresponding rewards for each specific event to train the agent is not conducive to generalization and is even impossible. Instead, we train the MCCAN to learn to do optimal event $E^*$ at location $L$. Thus, the $m=<L,E,T^{mc}>$ can be replacing by $m=<L,E^*,T^{mc}>$. Notably, $E^*$ is relative to each $L$ with respect to each agent. These meta-commands are broadcast to all agents and humans to communicate.
> > > > > >
> > > > > > During Communication, for agents, the MCC framework will also replace $E^H$ with $E^*$ when the agent receives a meta-command from humans. Then, the CS selects the optimal meta-command from the candidate set {${<L^H, E^*, T^{mc}>, <L, E^*, T^{mc}>}$}. For humans, the MCC framework will reverse the meta-commands $<L,E^*,T^{mc}>$ from agents into explicit messages and send them to humans through the in-game signaling system.
> > > > > >
> > > > > > After Communication, the MCC framework uses the MCCAN to perform including reaching $L$ and doing optimal event $E^*$ at $L$ based on the selected meta-command. And humans also select optimal meta-command and do corresponding optimal micro-actions based on their value systems.
> > > > > >
> > > > > > We have reorganized Section 3.2 and included these descriptions in Section 3.2 to remove possible ambiguity.
> > > > > >
> > > > > > [1] Yiming Gao, et al. "Learning diverse policies in moba games via macro-goals". NeurIPS'2021.
> > > > > >
> > > > > > ---
> > > > > >
> > > > > > Besides, we have revisited our paper and made a second revision to the paper to address clarity issues and make the paper more readable. Please see our [General Response](https://openreview.net/forum?id=q3F0UBAruO&noteId=mWbdwB6Nymt).
> > > > > >
> > > > > > Best regards,
> > > > > >
> > > > > > Paper 1657 Authors

---

### Official Review · Reviewer_peTd · 2022-10-29

**Confidence:** 4
**Correctness:** 3
**Technical Novelty And Significance:** 3
**Empirical Novelty And Significance:** 3
**Recommendation:** 6

**Clarity, Quality, Novelty And Reproducibility:**

Clarity: As explained above this is the biggest weakness of the paper. I found it quite difficult to understand.

Quality: This is somewhat hard to evaluate due to the lack of clarity though overall I found it to be high quality.

Originality / novelty: While interpretable human-AI communication has been studied before, this is (to my knowledge) the first case study with deep learning in a setting as complex as a MOBA game.

Reproducibility: Given the lack of clarity this paper would be hard to reproduce.

**Strength And Weaknesses:**

Strengths:
1. The paper tackles an important and significant problem domain – interpretable communication with humans in the service of human-AI collaboration.
2. The experimental results are strong, showing both improvements on win rate for the game, as well as improvements in human preference over which agents to play alongside.

Weaknesses:
1. The paper is hard to understand; despite poring over the details I would not know how to reproduce the paper’s results. See the list of questions below.
2. The authors claim that the MC-Base baseline (where meta-commands are not communicated) can be considered SOTA since it has the same level of performance as the SOTA WuKong agent, citing an appendix which is not present. It is not clear why this claim is true – did the authors evaluate the win rate between these agents? If so, why not also evaluate the win rate between MCC and WuKong, if you have access to the WuKong model? Even if you have evaluated the win rate of MC-Base vs WuKong, it is still good to compare MCC to WuKong, as chances of winning are not necessarily transitive (i.e. even if MCC beats MC-Base and MC-Base beats WuKong, it could still be that MCC loses to WuKong, though it is not likely).
3. The proposed method requires some domain-specific hardcoding (e.g. to define the message protocol, to define the set of possible events E, etc).

Notes on clarity and questions for the authors:

1. Which parts of my summary are incorrect?

2. In a message <L, E, T>, does the action E have to be selected from some predefined list of actions? If not, what are the legal values of E?

3. When describing the message <L, E, T>, in the example you suggest that T could be “until the dragon is killed”. However, in the experiments it sounds like T is always set to be some integer. Is it possible to have T be something like “until the dragon is killed” or does it have to be an integer? If it is possible, how is that implemented?

4. Is it possible for an agent to choose not to send any message?

5. In Figure 4(c), it appears that the AI agents must coordinate to send a single meta-command to the humans, which is done by using the meta-command selector π_ω(o, C). What do you use for the input o in this case, given that there are multiple AI observations but the CS only takes one observation?

6. I would assume that the CS should choose which meta-command to execute based on the probability of executing the meta-command successfully and the value of success for the rest of the game. This means that when training the CS, we need to give it a reward signal that captures how valuable it is to successfully complete the meta-command, which you would get from the environment rewards *after* the meta-command is complete. However, the CS optimization objective explicitly considers rewards only up to the end of the meta-command, rather than considering rewards after the completion of the meta-command. Why do you choose this instead of considering rewards up to the end of the game (or up to some fixed horizon length after the end of the meta-command)?

7. For the MCCAN training, the authors say

> We adopt an intrinsic reward r^{int}_t(s_t, m_t, s_{t+1}) $= |f^{ce}(s_t) − m_t| − |f^{ce}(s_{t+1}) − m_t|$ to guide the process of executing the meta-command $m_t$, where $f^{ce}$ is a hand-crafted command extraction function.

I am confused about what is happening here. Presumably this intrinsic reward somehow guides the policy to take actions that would achieve the meta-command $m_t$, since nothing else in the MCCAN training gives it that incentive. But how does it do this? My best guess is:

- We create $f^{ce}$, a hand-crafted function that “guesses” which meta-command is being executed based on the current state / observation
- We want the agent to take actions that cause $f^{ce}$ to make the correct “guess”, since that corresponds to taking actions that complete the meta-command, so we want the agent to minimize the distance between the guess from $f^{ce}$ and the actual meta-command at the end of the meta-command time interval, and which suggests a reward of $- |f^{ce}(s_{T^{mc}}) - m|$
- We apply potential shaping [1] over the time length of the meta-command to get the intrinsic reward above. (Though if this were the case there should be a discount factor in the reward.)

Is this correct? If so, why don’t you have the discount factor? Also, wouldn’t this incentivize the agent to complete the meta-command on the last timestep $T^{mc}$, rather than completing it as fast as possible?

8. For the MCCAN training, what is the training environment? The CS has not yet been trained so it cannot be used. Are you using a self-play setting where each of the 10 agents always executes its own meta-command generated by the CEN?

Other notes:

Section 4.4 (Collaborative Interpretability Analysis) would be improved by some quantitative metrics: for example, you could ask players to choose key moments during the game, and then report how often the Q-value ordering agrees with the expert player rank ordering.


[1] Ng, Andrew Y., Daishi Harada, and Stuart Russell. "Policy invariance under reward transformations: Theory and application to reward shaping." Icml. Vol. 99. 1999.

**Summary Of The Paper:**

I found this paper fairly hard to understand, so I will write a summary that’s fairly different from the author’s presentation that would have been easier for me to understand, and the authors can tell me if my summary is inaccurate.

There have been several agents that play multiplayer online battle arena (MOBA) games, such as OpenAI Five (Dota), AlphaStar (StarCraft), and the WuKong model (Honor of Kings). These agents play _competitively_ against humans: 5 copies of the agent control the 5 heroes on the team. However, we would also like to have AI agents _collaborate_ with humans: we can test this in MOBA games by having the team of 5 heroes be controlled by a mixture of humans and agents.

A key challenge in human-AI collaboration (HAC) is how to facilitate communication between the humans and the AIs. The core contribution of this paper is to propose such a method and demonstrate its utility in Honor of Kings.

First, we design a structured message format that humans can understand, which is a triple <L, E, T>. L is the location, which is one of 144 grid squares on the map. E is the task to perform, such as “kill the dragon”; I assume it is selected from a predesigned set of possible actions, though the paper does not say. T is an integer that specifies a number of timesteps in which to complete the action. The overall message <L, E, T> is to be interpreted as “in the next T timesteps, go to L and complete action E”. Only these instruction messages are supported; there is no other kind of communication.

Then, for a single timestep, our agents work as follows:

1. From their input observation o (and history, perhaps), generate a candidate message <L, E, T> to send to the other team players (whether AI or human). This is done by a Command Encoding Network (CEN) π_ϕ(m|o) (where m is the generated message / meta-commands).
2. Collect messages { <L, E, T> } from other agents. Select which of the messages to follow from amongst these messages and the message you generated yourself. This is done by a Meta-Command Selector (CS) π_ω(o, C) (where C is the set of received messages / meta-commands).
3. Select an action to execute based on the chosen message and the input observation. This is done by the Meta-Command Conditioned Action Network (MCCAN) π_θ(a|o, m), where m is the message / meta-command chosen in the previous step, and a is the action.

The human players play as normal, but are only shown one of the generated meta-commands. This is done by the CS but I am not clear how (see Q5 in my list of questions).

Training for these 3 models works as follows:

1. The CEN π_ϕ(m|o) is trained using supervised learning on an expert dataset that is automatically extracted from a dataset of expert gameplay using a hand-crafted function.
2. I don’t understand how the MCCAN is trained (see questions in the Strengths and Weaknesses section).
3. The CS is trained after the CEN and MCCAN are already trained. We simply use the full agents in a self-play setting, and train the CS using deep RL. Instead of using the environment reward function over the entire game, we instead use only the rewards during the execution of the current meta-command, and we weight the rewards obtained after reaching the location L twice as highly as the rewards obtained prior. (We also use multi-head value functions that predict different aspects of the reward, e.g. gold obtained vs. enemies killed, as done in prior work.)

The resulting MCC agent is compared against various ablations that show the value of communicating meta-commands between agents in an all-AI setting. The agent and various baselines are also paired with humans; the MCC agent achieves the highest winrate and is also most preferred by the human expert players.

**Summary Of The Review:**

I want to like this paper: it tackles an important problem, it proposes a high-level design that is relatively simple and has a clear reason why it would help, and it shows strong experimental results in a complex, high-dimensional setting.

However, as the paper is currently written, (a) it was quite difficult for me to understand even the high-level design, though I did eventually figure it out, and (b) even after that I cannot understand the details of how the method actually works. This is a key requirement for a scientific paper, and so unfortunately I would recommend against acceptance.

(Due to the lack of clarity I am also reducing my score for correctness, as there were some claims in the paper that I could not evaluate because the necessary information wasn’t present.)

I also have some other qualms (weaknesses 2 and 3 above) but they are minor in comparison and I would recommend acceptance if those were the only weaknesses.

UPDATE: After the author response many of the necessary details have been added, and so I am raising my score. However, I still find the paper relatively hard to understand.

---

> ### Author Response · Authors · 2022-11-14
> **Response to Reviewer peTd (1/2)**
>
> Thank you for carefully reviewing our paper! We provide clarification below for your questions and concerns. If you have any further questions or comments, we will be happy to discuss them further.
>
> ---
>
> **Clarification on the summary (Q1)**.
>
> **A1**: Overall, this summary can cover the motivation and process of the MCC framework. But there are some points of deviation that we need to clarify.
>
> > I assume $E$ is selected from a predesigned set of possible actions, though the paper does not say. **(Q2)**
>
> **A2**: We do not predesign a set of possible events for $E$, because this will rely on game-specific domain knowledge (e.g., specific rewards need to be set for guiding the agent to complete each $E$), and is not conducive to generality. To make meta-commands more general, we use extrinsic rewards (environmental rewards) to train the MCCAN to learn what event $E$ is optimal to do at location $L$, just as humans do optimally micro-operations at location $L$ based on their own value systems. We have supplemented these descriptions in Section 3.2.
>
> > The human players play as normal, but are only shown one of the generated meta-commands. This is done by the CS but I am not clear how. **(Q5)**
>
> **A5**: In Human-Agent Collaboration Test, to make the agent behave like humans (at most one human sends his/her meta-command at a time step), we restricted agents from sending their meta-commands, i.e., only one agent sends a valuable meta-command to human teammates from the agents' current meta-commands, as shown in Figure 4(c). Specifically, this process consists of several steps: (1) the MCC framework randomly chooses a human teammate (note that human and agent observations of the CS are shared); (2) the MCC framework uses his/her observation and all agents' meta-commands as the input of the CS, and obtained the estimated values of these meta-commands; (3) the MCC framework selects the meta-command with the highest value; (4) the MCC framework selects the corresponding agent to send the meta-command. The above process is executed with an interval of 20 seconds. Intuitively, the meta-command sent in this way is valuable to him/her, and he/she would be more willing to respond to it, so as to achieve the effective collaboration with agents. We have supplemented these descriptions in Appendix C.2.2.
>
> > I don't understand how the MCCAN is trained. **(Q7 & Q8)**
>
> **Optimization Objective (A7)**: We train the MCCAN based on the WuKong [1] model with the goal of achieving a high completion rate for meta-commands while ensuring that the win rate does not decrease. To achieve this, we introduce extrinsic rewards (including individual and team rewards, see in Table 4) and intrinsic rewards $r_{t}^{int}(s_t, m_t, s_{t+1}) = |f^{ce}(s_t) - m_t| - |f^{ce}(s_{t+1}) - m_t|$, where $f^{ce}$ extracts the agent's location from state $s_t$, and $|f^{ce}(s_t) - m_t|$ is the distance between the agent's location and the meta-command's location in time step $t$. Intuitively, the intrinsic rewards are adopted to guide the agent to the location $L$ of the meta-command and stay at $L$ to do some event $E$. The extrinsic rewards are adopted to guide the agent to perform optimal actions to reach $L$ and do optimal event at $L$. Besides, the extrinsic rewards are used to optimize the agent's performance to ensure that the win rate does not decrease. We use $\alpha$ to weigh the intrinsic rewards and the extrinsic rewards. The experimental results about the influence of $\alpha$ on the performance of the MCCAN are shown in Figure 16. We have supplemented these descriptions in Section 3.2 and Appendix B.6.3.
>
> **Training Environment (A8)**: We use a similar training environment as the WuKong agent [1] and the OpenAI Five agent [2], i.e., the agent interacts with the environment in a self-play manner. Specifically, for the MCCAN training, each of the 10 agents always executes its own meta-command generated by the CEN. And for each agent, every $T^{mc}$ time steps, the pre-trained CEN $\pi_\phi (m | o)$ generates a meta-command $m_t$ based on its observation $o_t$. The meta-command $m_t$ is then kept for $T^{mc}$ time steps and sent to the MCCAN continuously. During the interval $[t, t+T^{mc}]$, the MCCAN predicts a sequence of actions based on $o$ and $m_t$ for the agent to perform. We have supplemented these descriptions in Appendix B.6.3.

---

> > ### Author Response · Authors · 2022-11-14
> > **Response to Reviewer peTd (2/2)**
> >
> > **Q3**: Is it possible to have T be something like “until the dragon is killed” or does it have to be an integer?
> >
> > **A3**: Defining an exact execution time for each meta-command is not conducive to generality. We set $T^{mc}$ based on the statistics of the time taken by humans to complete the meta-command. Figure 14 shows that 80% of meta-commands can be completed within the time of 20 seconds in real games. Thus, $T^{mc}$ is set to 300 time steps (20 seconds).
> >
> > ---
> >
> > **Q4**: Is it possible for an agent to choose not to send any message?
> >
> > **A4**: Yes. To improve the generalization of CS to the number of meta-commands, we set a probability $p$ for agents to send their meta-commands in the Training Environment (Figure 4(a)). Thus, each agent has a probability $(1-p)$ of not sending its meta-command. Furthermore, in the Human-Agent Environment (Figure 4(c)), we also restrict at most one agent to send its meta-command at each time step. We can also set a threshold to filter out low-value meta-commands based on human’s value prior. Agents that are filtered out of meta-commands are prohibited from sending meta-commands.
> >
> > ---
> >
> > **Q6**: Clarification on the CS optimization objective.
> >
> > **A6**: We did not consider the reward for successfully completing the meta-command, mainly because this would introduce a bias to the CS optimization objective. For example, A is a meta-command that is farther from the agent, but more valuable. B is a meta-command that is very close to the agent but of little value. The agent has a higher probability of successfully executing B, but it may not help the game win. Thus, we only consider the extrinsic rewards (including individual and team rewards) to optimize and let the agent learn which meta-command to execute is more conducive to winning the game. In fact, we have considered rewards up to the end of the game. As we mentioned in Section3.3, the CS optimization objective is to maximize the expected discounted return $G_t^{mc}$, which includes the current meta-command execution reward $R^{mc}_t$ and all discounted meta-command execution rewards until the end of the game.
> >
> > ---
> >
> > **Other issues**
> >
> >
> > **Q1**: Even if you have evaluated the win rate of MC-Base vs WuKong, it is still good to compare MCC to WuKong, as chances of winning are not necessarily transitive.
> >
> > **A1**: Thank you so much for pointing this out. It is a very necessary result to make the conclusion more solid. To eliminate the concerns about the WR transitivity, we added additional experiments to compare the MCC agent and baseline agents against the WuKong (SOTA) agent for each 600 matches in Test I Environment. The WRs (P1 versus P2) are shown in the Table below:
> >
> > | P2\P1 | MC-Base | MC-Rand | MC-Rule | MCC |
> > | :---: | :---: | :---: | :---: | :---: |
> > | WuKong | 49.2% | 18.5% | 39.5% | 57.8% |
> >
> > As this Table shows, the effective CS mechanism in MCC agents enables the agents to communicate and collaborate effectively, which improves the WR compared with the WuKong agent (SOTA). We have included these results in Appendix B.6.4.
> >
> > ---
> >
> > **Q2**: The proposed method requires some domain-specific hardcoding (e.g. to define the message protocol, to define the set of possible events E, etc).
> >
> > **A2**: The meta-commands and MCC framework are generic to other popular MOBA games, e.g., Dota2 and League of Legends. Our future work is to design meta-commands into more general representations, such as natural language, and then apply them to other categories of games, such as First-Person Shooters (FPS) and Massively Multiplayer Online (MMO).
> >
> > ---
> >
> > [1] Deheng Ye, et al. "Towards playing full moba games with deep reinforcement learning". NeurIPS'2020.
> >
> > [2] OpenAI, et al. "Dota 2 with large scale deep reinforcement learning". arXiv'2019.

---

> ### Author Response · Authors · 2022-11-25
> **Looking forward to your reply**
>
> Dear Reviewer peTd,
>
> We thank you again for your endorsement of the motivation and the experimental studies of our work. We also appreciate you carefully reviewing our paper and raising the lack of clarification on the MCC framework might lead to difficulties in reading and biases in comprehension. We have responded in detail to all your concerns and included relevant clarifications to the paper and appendix. We believe that after including these clarifications, the revised paper would be more accessible and the quality can be reinforced.
>
> If you have any further questions or comments, we will be happy to discuss them further. We are looking forward to your feedback.
>
> Best regards,
>
> Paper 1657 Authors

---

> > ### Comment · Reviewer_peTd · 2022-11-27
> > **Thanks for the clarifications**
> >
> > These have answered most of my questions (and have been added to the paper) and so I will raise my score to be on the side of accepting the paper. However, I still find the paper to be quite dense and hard to understand, and so will raise to 6 rather than to 8.

---

> > > ### Author Response · Authors · 2022-11-28
> > > **Thanks for your feedback**
> > >
> > > Dear Reviewer peTd,
> > >
> > > Thanks again for your feedback and for raising your score. We will further polish the paper and move relatively unimportant content to the appendix to reduce the reading burden of the paper.
> > >
> > > Best regards,
> > >
> > > Paper 1657 Authors

---

### Official Review · Reviewer_Lfxb · 2022-11-03

**Confidence:** 3
**Correctness:** 3
**Technical Novelty And Significance:** 2
**Empirical Novelty And Significance:** 3
**Recommendation:** 6

**Clarity, Quality, Novelty And Reproducibility:**

Clarity: The description of the meta command framework is clear. The details of the compute requirements for the experiments, training time, batch size, and important hyperparameters are provided. They also point out the tricks used to enhance the training process of the command selector.

Quality: Although the idea of having a common ground in human-agent collaboration might not be new, its instantiation in MOBA games is innovative. The evaluations (both on agent-only and human subject teams) are well described even though reproducing the results might be a hassle due to the extensive computational requirements. Analysis on objective performance and subjective preferences was provided. The paper also addresses the potential limitations of the work and provides additional insights through ablation studies.


Novelty: This work, albeit being fiercely application-oriented, is novel in the sense that it shows the effectiveness of establishing a common ground in human-agent interactions instantiated in MOBA games. Even though the common grounding and techniques are specific to MOBA games, it could potentially inspire people in the AI community to have a human-interpretable communication framework for human-agent collaboration.


Reproducibility: Given the extensive use of compute resources, it might be difficult for many researchers to reproduce the results. Along with that, additional details on some of the blocks in each of the model structures (like the MLPs in CEN model structure or the LSTMs, CNNs in MCCAN structure) seem to be missing in order to replicate the results.


**Strength And Weaknesses:**

Strengths/Interesting aspects
1. Communication through a human-interpretable common ground: The MCC framework establishes a human interpretable communication framework where irrespective of what the agent’s internal representations are, the communication happens in a symbolic form which is interpretable to humans in the loop. This also inclines with having symbols as the lingua-franca [1] with the meta-commands being the symbolic information that is used as a common ground in human-agent interaction. The winrates from the online experiments show the effectiveness of the MCC framework with different types of agents and levels.

2. Human subject study for the performance of MCC teams: Even though the results show that as more humans are present in the team the win rate reduces against an all-agent SOTA method, in comparison to the other methods, human teams with MCC agents improve the overall team performance. Further, the response rates are also high in MCC-human teams.

3. Comparison of Command selector and high-level participant’s value systems - Although it doesn’t give a complete picture, it provides a preliminary insight into how the CS meta command selections are consistent with the ranking results

Weaknesses/Clarifications

1. Meta command is defined as a three element tuple where the event E (what to do ) is also part of it. The paper states the following “To achieve the former, a hand-crafted command converter function f cc is used to generate L of meta-commands by extracting the location from explicit messages, such as signals, sent by humans. To achieve the latter, we use a Command Encoding Network (CEN) πϕ(m|o) to generate L of meta-commands.” It is not clear as to how the event E in a meta command is incorporated (if needed) in the meta command execution if in the command conversion stage the output is just the locations extracted/generated from the explicit human commands/observations. Clarification on this aspect should be provided in the paper.

2. As much as it extends the SOTA in Honor of Kings, the common ground here is game-specific. The meta commands seem to be specifically designed for Honor of Kings. It is not clear if the meta commands or techniques are generic to other MOBA games either.


[1] Kambhampati, Subbarao, et al. "Symbols as a lingua franca for bridging human-ai chasm for explainable and advisable ai systems." Proceedings of the AAAI Conference on Artificial Intelligence. Vol. 36. No. 11. 2022.

**Summary Of The Paper:**

This work investigates human-agent collaboration in Multiplayer Online Battle Arena (MOBA) games and presents a human interpretable communication framework through which humans and agents can communicate their respective strategies during the game. It introduces the concept of meta commands which acts as the common ground for communication of macro-strategies between the humans and the agents. Further, the framework includes a meta command selector which estimates the value of the meta commands and selects a reasonable meta command to follow. The results show that using such a Meta Command Communication (MCC) framework improves the overall team performance and beats the current SOTA in MOBA games.

On a broader level, MOBA games present an interesting set of challenges for Game AI, one of which is human-agent collaboration. This paper addresses this challenge and shows that having an interpretable communication framework between humans and agents can achieve effective human-agent collaboration. They evaluate the MCC agent on agent-only environments and on online experiments with humans of different levels.

**Summary Of The Review:**

Overall, this work provides a novel instantiation of a human-interpretable common ground in MOBA games. The results show that such a framework improves human-agent collaboration and thereby the overall team performance. The framework on the other hand seems to be specific to the game used as the test-bed, Honor of Kings. But the promising objective and subjective evaluation of the framework could potentially inspire people in the AI community to have a human-interpretable communication framework for human-agent collaboration. Hence, this paper can be accepted to ICLR 2023.

---

> ### Author Response · Authors · 2022-11-14
> **Response to Reviewer Lfxb**
>
> Thank you for your thorough and constructive comments. We provide clarification below for your questions and concerns. If you have any further questions or comments, we will be happy to discuss them further.
>
> ---
>
> **Q1**: It is not clear as to how the event E in a meta-command is incorporated (if needed) in the meta-command execution.
>
> **A1**: We do not predesign a set of possible events for $E$, because this will rely on game-specific domain knowledge (Specific rewards need to be set for guiding the agent to complete each $E$), and is not conducive to generality. To make meta-commands more general, we use environmental rewards to train the MCCAN to learn what event $E$ is optimal to do at location $L$, just as humans do optimally micro-operations at location $L$ based on their own value systems. But, if there is a customized need for $E$, we can set $E$ to a predesigned event set, train a specific CEN, and use these predefined events and their corresponding customized rewards to guide the training of the MCCAN. We have supplemented these descriptions in Section 3.2.
>
> ---
>
> **Q2**: It is not clear if the meta commands or techniques are generic to other MOBA games.
>
> **A2**: The meta-commands and MCC framework are generic to other MOBA games, e.g., Dota2 and League of Legends. First, these MOBA games including HoK have similar game environments [1] (map layout, two areas, three lanes, four resource areas, etc.). Thus, the $L$, i.e., where to go, of meta-commands is generic to these MOBA games. Second, these MOBA games have similar mechanics [1], i.e., two hostile camps compete for resources through individual micro-operations and team collaboration on macro-strategies, and finally win the game by destroying the enemy's crystal. Thus, the $E$, i.e., what to do, of meta-commands is generic to these games. Third, although the time limit $T^{mc}$ of meta-commands may vary from game to game, we can still count the respective $T$ from human data in each game. Fourth, these MOBA games provide an in-game signaling system [1], which naturally supports the communication between agents and humans via the generically defined communication protocol, i.e., the meta-command. Thus, meta-commands can be regarded as the **lingua franca** [2] of MOBA games, and the MCC framework can be easily applied to other MOBA games. Our future work is to design meta-commands into more general representations, such as natural language, and then apply them to other categories of games, such as First-Person Shooters (FPS) and Massively Multiplayer Online (MMO).
>
> ---
>
> **Q3**: Provide additional details on the CEN, MCCAN, and CS model structures.
>
> **A3**: We have released the MCC framework code and the detailed parameter settings, please see at https://github.com/ICLR2023-MCC/MCC.
>
> ---
>
> [1] do Nascimento Silva, Victor, and Luiz Chaimowicz. "Moba: A new arena for game ai". arXiv'2017.
>
> [2] Kambhampati, Subbarao, et al. "Symbols as a lingua franca for bridging human-ai chasm for explainable and advisable ai systems". AAAI'2022.

---

> ### Author Response · Authors · 2022-11-25
> **Looking forward to your reply**
>
> Dear Reviewer Lfxb,
>
> We appreciate your positive comments on the novelty and experimental studies of our work. We have responded in detail to all your questions and concerns and included relevant clarifications to the paper and appendix. If you have any further questions or comments, we will be happy to discuss them further. We are looking forward to your feedback.
>
> Best regards,
>
> Paper 1657 Authors

---

### Author Response · Authors · 2022-11-18
**General Response: Summary of Paper Updates**

Dear Reviewers and Area Chairs,

We thank all reviewers for carefully reviewing our paper and providing valuable suggestions. We greatly appreciate your feedback. Please see below our general responses and summary of the revised version.

**(I) Clarification.** We have made further and more detailed clarifications on the MCC framework including the Meta-Command, the CEN, the MCCAN, and the CS.

**(II) Reproducibility.** We have supplemented reproducibility details including network structures, detailed parameter settings and released the MCC framework code at https://github.com/ICLR2023-MCC/MCC.

We believe that after revising the paper according to all reviewers' suggestions, the quality of this work can be improved. If you have any further questions or comments, please post them and we will be happy to have further discussions. We are looking forward to your feedback.

Best regards,

Paper 1657 Authors

---

> ### Author Response · Authors · 2022-11-29
> **General Response: Summary of the Second Revision**
>
> Dear ALL Reviewers,
>
> We greatly appreciate your further replies and suggestions. Based on your latest valuable comments, we have revisited our paper and made a second revision to the paper to address clarity issues and make the paper more readable. Please see the revised paper at https://raw.githubusercontent.com/ICLR2023-MCC/MCC/main/paper/MCC_revision_V2.pdf. Relevant modifications are marked in red and are summarized as follows.
>
> - **Simplify Figure 2 (The MCC framework)**: We decouple the temporal process and the working mechanism at each stage, and reduce the symbols that may cause ambiguity.
>
> - **Reorganize Section 3.1 (Overview of the MCC framework)**: Based on the new Figure 2, we describe the working mechanism of each stage in the MCC framework, and clarify the execution time step of each stage to reduce the ambiguity at the time level.
>
> - **Reorganize Section 3.2 (Meta-Command Conversion & Execution)**: We have included detailed descriptions of the definitions and conversions about $L$, $E$, and $T$. We have also included a clear introduction to the mechanism of CEN and MCCAN.
>
> - **Simplify Figure 3(a) (The training process of MCC)**: We simplify the symbols in the figure for a clearer presentation.
>
> - **Reorganize Section 3.3 (CS Optimization Objective)**: We reorganize the optimization objective of CS to be more clear and eliminate possible ambiguity.
>
> - **Polish the paper**.
>
> We hope that the new revised version could address the remaining clarity issues. If you have any further questions or comments, we will be happy to have further discussions.
>
> Best regards,
>
> Paper 1657 Authors

---

### Decision · Program_Chairs · 2023-01-20

**Decision:**

Accept: notable-top-25%

**Justification For Why Not Higher Score:**

Method is specific to MOBA games so insights may not immediately generalise to playing other games or non-gaming interactions with human users.

**Justification For Why Not Lower Score:**

Significant large-scale result likely of broad interest due to its contributions to both gameplaying AI and natural language interactions

**Metareview: Summary, Strengths And Weaknesses:**

This paper demonstrates a method to enable human-agent communication can improve win-rate and subjective preference when playing the MOBA Honor of Kings. The method is only applicable to MOBA games, but as one of the most popular modern genres this could already enable applications enjoyed by millions of people worldwide. Other concerns raised by reviewers were predominately related to clarity of the paper and reproducibility of the work. The authors responses helped reduce these concerns, but this feedback should be carefully considered in further updates to the paper and in any future presentations. The core contribution of an agent that mixes both gameplaying abilities and grounded communication with humans is likely of broad interest to the ICLR community.

**Note From Pc:**

if the above contains the word "oral" or "spotlight" please see: "oral" presentation means -> notable-top-5% and "spotlight" means -> notable-top-25%. As stated in our emails, we are disassociating presentation type from AC recommendations